# Data Mixing Laws: Optimizing Data Mixtures by Predicting Language Modeling Performance

**Jiasheng Ye**[1*], **Peiju Liu**[1*], **Tianxiang Sun**[1], **Jun Zhan**[1], **Yunhua Zhou**[2†], **Xipeng Qiu**[1†]
[1]Fudan University, [2]Shanghai AI Labortory
{jsye23,pjliu23}@m.fudan.edu.cn
zhouyunhua@pjlab.org.cn xpqiu@fudan.edu.cn

## Abstract

Pretraining data of large language models composes multiple domains (e.g., web texts, academic papers, codes), whose mixture proportions crucially impact the competence of outcome models. While existing endeavors rely on heuristics or qualitative strategies to tune the proportions, we discover the quantitative predictability of model performance regarding the mixture proportions in function forms, which we refer to as the *data mixing laws*. Fitting such functions on sample mixtures unveils model performance on unseen mixtures before actual runs, thus guiding the selection of an ideal data mixture. Furthermore, we propose nested use of the scaling laws of training steps, model sizes, and our data mixing laws to predict the performance of large models trained on massive data under various mixtures with only small-scale training. Experimental results verify that our method effectively optimizes the training mixture of a 1B model trained for 100B tokens in RedPajama, reaching a performance comparable to the one trained for 48% more steps on the default mixture. Extending the application of data mixing laws to continual training accurately predicts the critical mixture proportion that avoids catastrophic forgetting and outlooks the potential for dynamic data schedules.[1]

## 1 Introduction

Pretraining data for large language models (LLMs) are typically a mixture of multiple domains, varying from English to minority languages (Doddapaneni et al., 2021; Li et al., 2023), from casual dialogs to formal academic writings (Taylor et al., 2022), and from texts to modalities like images and speeches (Zhan et al., 2024), among others. These data interplay with each other, showing complex relationships including facilitation, being unrelated, or conflict (Guo et al., 2024). This necessitates adjusting the mixture proportions of training data to balance the model capabilities while harnessing synergies across domains, thus enhancing the competence of the outcome models, as highlighted by extensive practices (Gururangan et al., 2020; Zhou et al., 2023; Xie et al., 2024a; Fan et al., 2024).

Nonetheless, it remains elusive to figure out an ideal training data mixture. Most existing practices tune the mixture through heuristics to upsample a proportion of high-quality or underrepresented data without disclosing the concrete criteria in detail (Gao et al., 2020; Touvron et al., 2023a; Bai et al., 2023; Bi et al., 2024). While some studies propose automatic algorithms to qualitatively optimize data mixture (Xie et al., 2024a; Fan et al., 2024), it is hard to predate the effect of these strategies before the actual training run. In contrast, encouraged by advances in scaling laws that show model losses on a given set of evaluation data are quantitatively predictable for a wide range of variables (Kaplan et al., 2020; Hoffmann et al., 2022), we wonder whether this also holds for mixture proportions, so that ***we can estimate the outcome model performance given any mixture before actually training on them, including the desired one that reaches minimum loss.***

In this paper, we answer this proposition affirmatively. The intuition is that predicting the performance of unseen data mixture only involves interpolating among seen mixtures because the proportions are bounded between 0 and 1. For this reason, numerous function forms can lead to descent prediction accuracy, among which we try to find a simple one. In particular, we find that, given a mixture of $M$

---

[*]Equal contribution.
[†]Corresponding authors.

[1]Codes and data are available at: https://github.com/yegcjs/mixinglaws.

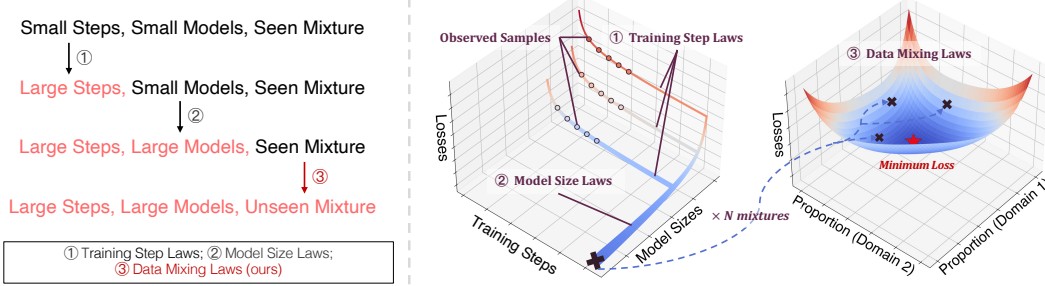

Figure 1: Illustration on our pipeline to optimize data mixture. **Left:** Our pipeline takes three steps. Starting from small-scale training results, the three steps use the scaling laws of training steps, model sizes, and data mixing laws to predict model performance on large steps, large models, and unseen mixtures, respectively. **Right:** Visualization of the three-step pipeline to predict model performance on the target model size, training step, and mixtures.

domains, an exponential function over the linear combination of the proportions, i.e.,

$$L_i(r_{1...M}) = c_i + k_i \exp\left(\sum_{j=1}^{M} t_{ij} r_j\right), \tag{1}$$

can predict the validation loss $L_i$ on any of the training domains $i$ accurately under a fixed model size and amount of training data, where $r_{1...M}$ are the proportions of the $M$ domains and $c_i, k_i, t_{ij}$ are parameters to fit. Fitting such functions on all the evaluated domains and calculating the weighted sum according to their proportions in the validation data leads to the prediction of final validation loss. Further, treating the validation proportions as learnable parameters allows fitting the estimated losses on a validation set end-to-end without explicitly decomposing it into known domains.

Despite the predictability, fitting the function between mixture proportions and validation loss, or the *data mixing laws* for simplicity, requires samples of numerous runs with different mixtures. Running these experiments with the same model size and the amount of training data as the target model is unreasonably expensive. Fortunately, fruitful research on scaling laws demonstrates impressive results that fitting power laws with small models and small data effectively predicts the losses on larger models and data over orders of magnitudes (Kaplan et al., 2020; Henighan et al., 2020; Hoffmann et al., 2022; Alabdulmohsin et al., 2022; OpenAI, 2023; Muennighoff et al., 2024; Bi et al., 2024). On this basis, we propose a pipeline to nested utilize the scaling laws of training steps, model sizes, and our data mixing law, so that we can study the impact of mixture proportions for the target model sizes and data amount with only experiments at the affordable scales, illustrated in Fig. 1.

Experimental results verify the reliability of our data mixing law and prediction pipeline. By predicting the overall validation loss, we optimize the training mixture of RedPajama for a 1B model trained on 100B tokens and achieve performance comparable to a model trained on default mixture for 48% more steps. The prediction on domain-specific validation sets also offers plausible references to the balance of model capabilities. Further applying our data mixing law to continual pretraining can accurately find the proportion that avoids catastrophic forgetting (French, 1999; Kirkpatrick et al., 2017; Luo et al., 2023), revealing the prospect of applying data mixing laws to guide a multi-stage pertaining, and thus a dynamic data schedule.

Overall, our contributions and findings are as follows:

- We discover the quantitative predictability of model performance regarding data mixture, and summarize this into a functional relationship, namely the data mixing laws.
- We propose a pipeline to predict model performance of large-scale training on different mixture proportions but only experiments on small models with few training data through nested use of scaling laws of training steps, model sizes, and data mixing laws.
- We experiment to verify the reliability of our data mixing laws and prediction pipeline, showing its effectiveness in optimizing model performance, balancing model capabilities, and the prospects of guiding the design of the data schedule.

## 2 BACKGROUND

We briefly review the pretraining process of large language models and summarize key findings from neural scaling laws, then we formalize the problem we study. Further related works are in App. A.

**Pretraining large language models.** We consider the task of pretraining an autoregressive language model $p_\theta$ via next-token predictions (Radford et al., 2018). The training dataset $\mathcal{D}_{\text{train}} = \{\mathcal{D}_i\}_{i=1}^M$ composes $M$ domains with mixture proportions $r \in \Delta^{M-1}$. In each training step, the task first samples a batch of domain indices according to the mixture proportions and then sample sequences of $L$ tokens from the sampled domains. Using the sampled data, it learns to optimize the negative log-likelihood of sampled data, i.e.,

$$\mathcal{L}_\theta = \mathbb{E}_{i \sim r, x_{0...L} \sim D_i} \left[ -\sum_{j=1}^L \log P_\theta(x_j | x_{0...j-1}) \right]. \tag{2}$$

To evaluate the learned model, we compute the loss on validation data $\mathcal{D}_{\text{val}}$.

**Scaling laws.** For a wide range of factors $x$, scaling laws (Kaplan et al., 2020; Henighan et al., 2020; Hoffmann et al., 2022) show that their effect on the losses $L$ follows power laws

$$L = c + kx^\alpha, \tag{3}$$

where $c$, $k$, and $\alpha$ are parameters to fit and $x$ can be model sizes, numbers of training data, training steps, and the amount of computation. Previous experience (Alabdulmohsin et al., 2022; OpenAI, 2023; Bi et al., 2024; Su et al., 2024) highlights the impressive predictability of scaling laws. Specifically, Eqn. 3 fitted on a collection of small models, training data, or computation can extrapolate to precisely predict the test loss of larger cases over orders of magnitudes. This enables practitioners to estimate the performance of a pretrained large language model without actually finishing the expensive runs. Recent development further shows various functional relationships between the performance of language models and a broader range of factors, including transfer learning (Hernandez et al., 2021), sparse architectures (Frantar et al., 2023), and repeated data (Muennighoff et al., 2024), consolidating the predictability of language model performance.

**Problem formalization.** We study optimizing the mixture proportions of pretraining data for large language models. Motivated by the impressive predictability of existing scaling laws, we try to tackle mixture optimization by establishing a quantitative framework that predicts the loss given any mixture proportion. Formally, for a training dataset comprising $M$ domains, we parameterize the function

$$L = f_\theta(r), \tag{4}$$

under the fixed model sizes and number of training steps, where $r = r_{1...M}$ is the proportion of the $M$ domains. Harnessing this function, we seek a mixture that achieves the desired performance. Without loss of generality, we search for the mixture that reaches minimum validation loss, i.e.,

$$r^* = \arg\min_r f_\theta(r). \tag{5}$$

## 3 THE PROPORTIONS OF DATA MIXTURES INFLUENCE MODEL LOSSES IN A QUANTITATIVELY PREDICTABLE WAY

In this section, we present our findings on the predictability of model losses regarding data mixtures, which boils down to functional relationships we refer to as the data mixing laws.

To discover the data mixing laws, we encounter two challenges posed by their characteristics.

 (i) *Multi-variables.* For a data mixing law for $K$ domains, we should consider $K-1$ degrees of freedom in the mixture proportions and, correspondingly, $K-1$ variables in the target function. The increase of variables considerably enlarges the scope of potential functions thereby complicating the identification of the function form.

 (ii) *Nonmonotonicity.* A monotonic relationship between losses and the proportion of any domain indicates that a lopsided mixture can achieve minimum loss without endeavors to balance domain proportions, which contradicts the practice. Therefore, differing from existing scaling laws that loss monotonically decreases with the scale of concerning factors, the data mixing law we study should accommodate non-monotonic functions. This nonmonotonic nature adds another layer of complexity to our analysis.

To navigate these challenges, we initially simplify the problem by studying a scenario where the relationship between loss and mixture proportions fits into a univariate monotonic function then retracts the simplifications progressively. In specific, we begin our study on the case where we only train on two domains thus avoiding multi-variables, and only consider the validation data coming

from one of the training domains to circumvent the non-monotonicity (Sec. 3.1). Subsequently, we broaden our framework to encompass training on multiple domains (Sec. 3.2) and explore the predictability of losses on general validation data that also comprises various domains (Sec. 3.3).

## 3.1 PILOT STUDY ON DOMAIN LOSSES UNDER TWO-DOMAIN MIXTURES

We begin our exploration with the simplest case where we only learn on mixtures of two domains and evaluate our model on the two domains respectively.

**Setups** We train 70M and 160M language models on the mixture of Github and Pile-CC subset from the Pile dataset (Gao et al., 2020) with five different mixture proportions, which are {0.25, 0.375, 0.5, 0.625, 0.75} for Github. We train all models with a batch size of 1M tokens for 30k steps, which is 30B tokens in total, and evaluate checkpoints at different steps on the validation set of GitHub and Pile-CC.

**Findings** Results in Fig. 2 reveal the quantitative predictability of domain losses given the domain proportions. We encouragingly find that, for checkpoints with the same size and trained with the same number of steps, after subtracting a shared constant[2], their domain losses in the log scale demonstrate a linear relationship to the domain proportion. This holds for both domains in our experiments. The result indicates that with other factors fixed, the domain losses of a pretrained language model regarding the domain proportion precisely fit into an exponential law[3]

$$L_i(r_i) = c_i + k_i \exp(t_{ii} r_i), \qquad (6)$$

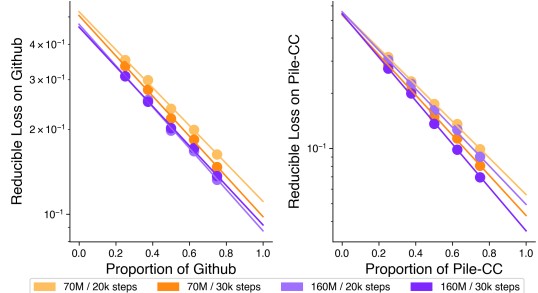

Figure 2: Quantitative predictability of domain losses on two domains, which are Github and Pile-CC. We train on the mixtures of these two domains and validate the outcome models on them separately. We train 70M and 160M models on five different mixtures of Github and Pile-CC and obtain the reducible losses by subtracting the original losses with a constant shared across models of the same sizes and trained for the same number of steps. The reducible losses in log scale show linear correlations to the domain proportions.

where $L_i$ and $r_i$ are validation loss and training mixture proportion of domain $i$, respectively, while $c_i$, $k_i$, and $t_{ii}$ are learnable parameters [4].

## 3.2 EXTENSION TO DOMAIN LOSSES TRAINED ON MULTI-DOMAIN MIXTURES

To accommodate real-world pretraining data that mostly contains more than two domains, we extend our investigation into multiple domains. For simplicity and the ease of visual aids, we start with the case of three domains.

**Setups** We train on the mixtures of GitHub, Pile-CC, and Books3 subset from the Pile for a total of 30B tokens and evaluate the model on the three domains, respectively. For specific mixtures, we grid search from $\{0, 0.125, 0.25, \ldots, 0.875, 1\}^3$ and retain valid ones in which three proportions sum up to 1 and do not use up all tokens in any of the domains, which results in 32 mixtures in total.

We utilize the losses on these experimented mixtures to identify the function forms between losses and mixture proportions through conjecture and then verification. In specific, we base our conjecture of possible forms on the following two principles.

- *Compatibility.* The form can reduce to Eqn. 6 if the number of domains $M = 2$.
- *Symmetry.* Any exchanging of variables should not change the functional form as we should not incorporate any domain-specific bias.

---

[2]The constant term, known as irreducible loss, arises from finite training data and the entropy of the evaluation data theoretically (Bishop, 2006; Henighan et al., 2020).

[3]While power laws are more common in existing studies on scaling laws (Kaplan et al., 2020; Hoffmann et al., 2022), we do not consider it for its ill-posed properties that the function value blows up when the variable, mixture proportion in our case, approaches 0. This contradicts the observations that the losses remain low (e.g., no more than 10) with the generalization between domains.

[4]Despite a simple case, our findings on two domains have practical applications to continue pretraining (Gururangan et al., 2020), where we aim to enhance a pretrained model on a given domain by training it on a mixture of the original pretraining data and upcoming domain data. Please see Sec. 5 for details.

Table 1: Mean absolute errors (MAE) of different candidate functions for predicting the target domain losses. We also include random guesses that randomly predict between the maximum and minimum loss of the training samples for reference. In specific, we report the MAE of the expectation of this random guess which predicts the median of the maximum and minimum loss. The training data contain $M = 3$ domains and we fit each function with the same 24 mixtures and validate on 8 other mixtures. The split is random. The lowest error for each target domain are in **bold** while the second lowest are underlined.

| Method | # Coeff. | GitHub | | Books3 | | Pile-CC | |
|--------|----------|--------|------------|--------|------------|--------|------------|
| | | Train | Validation | Train | Validation | Train | Validation |
| Random | - | 0.8895 | 0.8758 | 0.1291 | 0.1331 | 0.0768 | 0.1045 |
| M1 | 2M+1 | **0.0292** | **0.0312** | 0.0082 | 0.0121 | 0.0045 | **0.0050** |
| M2 | M+2 | 0.1558 | 0.3327 | 0.0114 | 0.0119 | 0.0072 | 0.0083 |
| M3 | M+2 | 0.3389 | 0.2177 | 0.0914 | 0.0465 | 0.0746 | 0.0947 |
| M4 | M+2 | 0.0298 | 0.0365 | **0.0062** | **0.0074** | **0.0036** | 0.0078 |

Together, the two principles lead to candidate functions that replicate the exponential term in Eqn. 6 for each training domain and combine them through operations that adhere to commutative law.

According to the two principles, we experiment with the following candidate functions:

$$\text{M1:}\quad L_i(\boldsymbol{r}) = c_i + \sum_{j=1}^{M} \left[ k_{ij} \exp\left(t_{ij} r_j\right) \right], \quad \text{M2:}\quad L_i(\boldsymbol{r}) = c_i + k_i \sum_{j=1}^{M} \exp\left(t_{ij} r_j\right),$$

$$\text{M3:}\quad L_i(\boldsymbol{r}) = c_i + k_i \exp\left( \prod_{j=1}^{M} t_{ij} r_j \right), \quad \text{M4:}\quad L_i(\boldsymbol{r}) = c_i + k_i \exp\left( \sum_{j=1}^{M} t_{ij} r_j \right).$$

We summarize their fitting errors on three target domains in Tab. 1.

**Findings** The results in Tab. 1 suggests both M1 and M4 gives reliable predictions while M4 has fewer coefficients. Therefore we adopt M4

$$L_i(r_{1\ldots M}) = c_i + k_i \exp\left( \sum_{j=1}^{M} t_{ij} r_j \right) \quad (7)$$

as the function forms of our data mixing law, where $L_i$ is the validation loss on $i$-th validation domain, $r_j$ is the proportion of the $j$-th training domain, and $c_i, k_i, t_{ij}$ are learnable parameters. The fitting results are in Fig. 5 and Fig. 3 demonstrates the prediction accuracy. The results indicate that Eqn. 7 fits the given samples well and estimates the unseen ones accurately.

**Meanings of the coefficients.** To provide more intuition, we discuss the meanings of the coefficients in Eqn. 7. In general, $k_i > 0$, thus the exponential term is always positive and the prediction loss is strictly greater than the constant $c$. Hereby, $c_i$ represents losses that are not reducible by adjusting the data mixture. $t_{ij}$, depending on both training domain $i$ and valida-

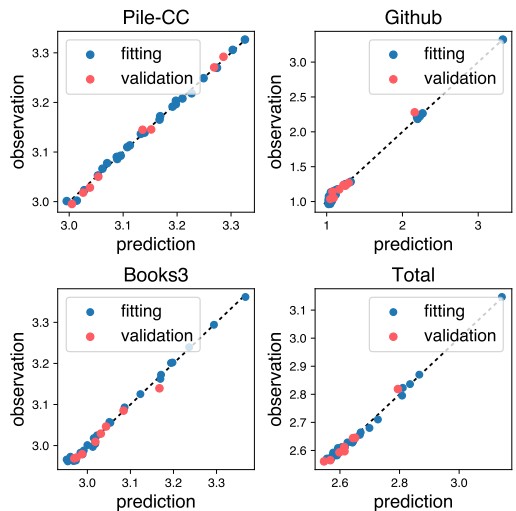

Figure 3: Prediction results on the domain losses and overall losses in the three-domain experiment. The domain losses are fitted with Eqn. 7 and we obtain the total losses through explicit domain aggregation of Eqn. 8.

tion domain $j$, shows the interaction between them. A negative $t_{ij}$ indicates that training data of domain $j$ helps reduce validation loss on domain $i$ and vice versa.

**Patterns of the coefficients.** We visualize normalized $t_{ij}$ of training and validating on the 5 domains mixture of the Pile[5] in Fig. 4. The relationship between domains can be categorized into 3 types.

---

[5]The Pile contains 22 fine-grained domains which are collected into five coarse-grained domains, i.e., academic, internet, prose, dialogues, and misc, where misc include Github and the DeepMind Mathematics

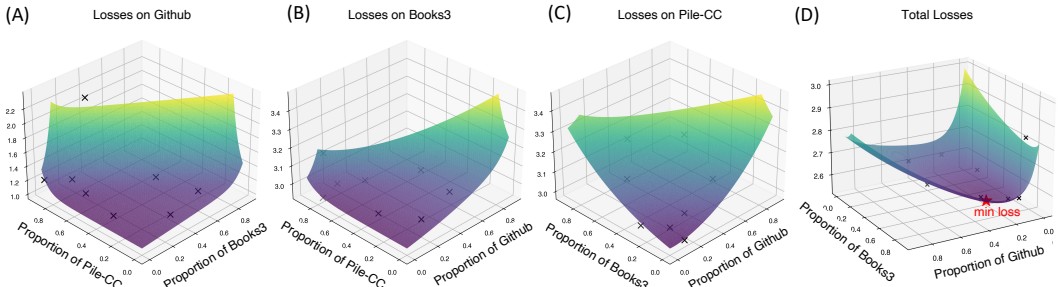

Figure 5: Quantitative predictability of domain losses on three domain mixtures, Github, Books3, and Pile-CC. We train on the mixture of these three domains and validate the outcome models on them as well. The surfaces show the predicted losses on (A) Github; (B) Books3; (C) Pile-CC; and (D) the overall validation set mixed with the three domains. ×: validation samples. ⋆: the predicted minimum loss on the overall validation set.

**Being unrelated:** The figure shows a highly sparse pattern where most of the domains have little relationship to each other and the validation performance of a domain is dominated by training data of the same domain, which supports the intuitive progressive mixture tuning strategy that adds data for underperforming capability during training (Team, 2023). Meanwhile, we also observe **facilitation** (e.g., training dialogue for the internet) and **conflict** (e.g., training symbolic data for prose) between domains, which indicates the room for leveraging domain interaction to enhance model performance.

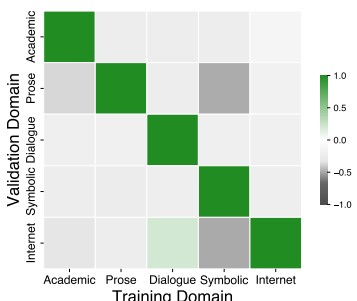

### 3.3 PREDICTING LANGUAGE MODELING PERFORMANCE OF ANY VALIDATION MIXTURE

We further loosen constraints in Sec. 3.1 and Sec. 3.2 that the validation data are from one of the training domains. We first consider the validation set to be a known composition of the training domains and then free this requirement for more general cases of arbitrary validation sets. These correspond to the two strategies we fit the data mixing laws, which we elaborate on as follows.

Figure 4: The interaction between different training and validation domains on the Pile. Each boxes are fitted normalized $t_{ij}$ from Eqn. 7. We normalize the value by $t_{ij}$ with the maximum absolute value for each validation set $i$ (i.e., $t_{ij}/t_{i,\arg\max_j |t_{ij}|}$), to compare the values intuitively. A larger value (greener) indicates the training domain helps learn the validation domain more.

**Explicit domain aggregation.** Considering a validation set made up of $K$ domains with the proportions as $s_{1\ldots K}$, the validation loss can be written into the weighted sum of domain losses. Thanks to the discovery of Eqn. 7, we can apply the equation to predict domain losses herein given a training mixture. Therefore, the functional relationship of the overall validation loss to the training mixture proportions expands into

$$L(r_{1\ldots M}) = \sum_{i=1}^{K} s_i L_i(r_{1\ldots M}) = \sum_{i=1}^{K} s_i \left[ c_i + k_i \exp \left( \sum_{j=1}^{M} t_{ij} r_j \right) \right]. \tag{8}$$

Using Eqn. 8, we can fit the loss on each validation domain $L_i$ and sum them up to obtain the prediction of overall loss.

**Implicit domain aggregation.** A remaining limitation is that we still need to acquire the components of validation data $s_{1\ldots K}$ in advance. This can be inconvenient if the validation set is collected separately from the training ones. For instance, the validation data may come from real-world user queries that cover unknown compositions of various domains. To remove the constraint on validation components, we assume that we can decompose the validation data into $K$ implicit domains whose losses are predictable with Eqn. 7, and we treat their proportions in the validation data $s_{1\ldots K}$ as

---

Dataset which are symbolic content. We do not experiment on fine-grained domains for their limited number of tokens available.

learnable parameters, leading to the final form of our data mixing laws[6]. With this perspective, we fit a data mixing law with the overall losses end to end.

Introducing implicit domains may draw concerns about the number of fitting samples exploding with the number of parameters to fit and questions on deciding the number of implicit domains without knowing the oracle. We study and discuss their impact, respectively.

***Do we need quadratic number of samples to fit Eqn. 8 as the number of domain grows?*** **No.** The parameters in Eq.8 scale as $\mathcal{O}(M \times K)$, where $M$ and $K$ represent training and implicit validation domains. Nevertheless, as shown in Fig.6, the quadratic growth in the number of parameters does not translate to quadratic growth in sample requirements. We attribute this to the high sparsity in the parameters as fig.4 reveals, which allows us to fit the equation with substantially fewer samples when using appropriate regularization. While using more samples decreases prediction errors, the number of samples that reach a similar precision level does not grow dramatically. This may pave the way for

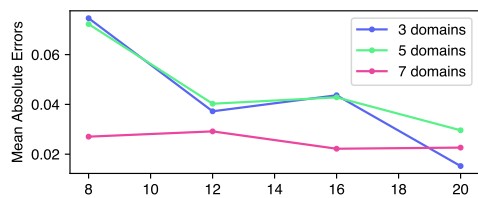

Figure 6: The mean absolute validation errors of Eqn. 8 fitted with different numbers of samples for training mixtures containing different numbers of training domains. For each number, we resample and select the batch of fitting mixtures that reach lowest errors.

applying implicit domain aggregations for cases with more training domains. Although concluding the exact number of samples required can be challenging due to the differences among training data, we can tune the fitting mixtures on the smallest experimented models, which is cheap and works well in practice (see Sec. 4.2 and App. D.3).

***How to choose the number of implicit domains?*** **Set it larger than the oracle one.** Fig. 7 shows our experiments where we train language models on the 5 coarse-grained domains of Pile and evaluate a validation set mixed with these 5 domains. We compare the errors obtained by implicit domain aggregation with different numbers of implicit domains to those obtained by explicit domain aggregation. We find that applying implicit domain aggregation and setting the number of implicit domains no smaller than the actual one (5 in the experimented case) results in lower errors than explicit domain aggregation. Moreover, the error remains low as we set the number of implicit domains much larger. This verifies the prediction accuracy of our implicit domain aggregation strategy for data mixing law and the number of implicit domains $K$ can be a large number without careful tuning[7].

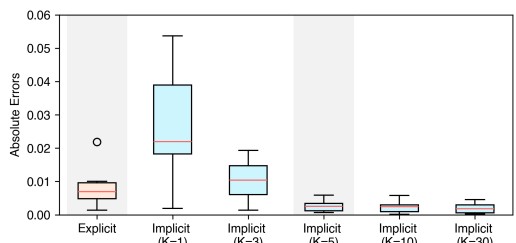

Figure 7: Prediction errors of the five-domain data mixing laws fitted with explicit and implicit domain aggregation. *Explicit domain aggregation*: we fit Eqn. 7 for five domains respectively and sum them up according to their weight in the overall validation sets. *Implicit domain aggregation*: we fit the losses on overall validation with Eqn. 8, assuming different numbers of implicit domains $K$ and treating the proportion of different implicit domains as learnable parameters.

# 4 NESTED SCALING LAWS PREDICT LOSSES TRAINED ON VARIOUS MIXTURES USING ONLY SMALL-SCALE EXPERIMENTS

## 4.1 A PIPELINE FOR LOSS PREDICTIONS

While data mixing laws enable us to predict the performance of models trained on unseen mixtures, fitting the laws directly on target scales is unaffordably expensive. Firstly, fitting the laws involves training multiple models across diverse mixtures with model sizes and token counts identical to the

---

[6]We note that the final forms of our data mixing law resemble a multilayer perception (see the computation graph Fig. 14). We include further discussion and implementation details in Appendix E.

[7]We set $K = 30$ if not stated in later experiments.

---

**Algorithm 1** A pipeline to predict losses of different mixture proportions on large models trained on massive data through small-scale training

---

**Input:** Validation data $D_{val}$, training data of $M$ domains $\{D_m\}_{m=1}^M$, target training steps $S_{target}$, target model size $N_{target}$, target mixture to predict $\boldsymbol{r}_{target}$, training steps to fit the step laws $S_0$, model sizes to fit the size laws $\{N_j\}_{j=1}^K$, and $N$ data mixtures $\{\boldsymbol{r}_i\}_{i=1}^N$ to fit.

**Output:** The validation loss of a $N_{target}$ model trained for $S_{target}$ steps on mixture $\boldsymbol{r}_{target}$, i.e., $L(N_{target}, S_{target}, \boldsymbol{r}_{target})$.

1: **for** Each mixture $\boldsymbol{r}_i$ **do**
2:     **for** Each model size $N_j$ **do**
3:         Train model of size $N_j$ on mixture $\boldsymbol{r}_i$ for $S_0$ steps to obtain $L(N_j, S < S_0, \boldsymbol{r}_i)$
4:         Fit training step scaling law $L(S)$ with $L(N_j, S < S_0, \boldsymbol{r}_i)$
5:         Predict $L(N_j, S_{target}, \boldsymbol{r}_i) \leftarrow L(S = S_{target})$
6:     **end for**
7:     Fit model size scaling law $L(N)$ with $L(N_{1...K}, S_{target}, \boldsymbol{r}_i)$
8:     Predict $L(N_{target}, S_{target}, \boldsymbol{r}_i) \leftarrow L(N = N_{target})$
9: **end for**
10: Fit the data mixing law $L(\boldsymbol{r})$ with $L(N_{target}, S_{target}, \boldsymbol{r}_{1...N})$
11: predict $L(N_{target}, S_{target}, \boldsymbol{r}_{target}) \leftarrow L(\boldsymbol{r}_{target}$

---

target ones. Furthermore, we must repeat this process for each target model size and training dataset[8]. Such expensive costs hinder the practical value of our data mixing laws.

We thus wonder whether we can obtain the losses of different mixture proportions without training at large scales. Fortunately, this idea gains endorsement from existing experiences that verify the impressive extrapolation of scaling laws of training steps and model sizes. In particular, OpenAI (2023) predicts the loss of the target model with merely $1,000\times-10,000\times$ less compute. Consequently, we can train small models with few training steps on different mixtures, and fitting scaling laws on them to estimate the losses of the target model size and the target number of training steps. We can then use the predicted losses to fit a data mixing law and search for the optimal mixture.

We illustrate the proposed pipeline in Fig. 1 with details depicted in Alg. 1. Scaling laws in our pipeline are largely based on the function forms of Chinchilla Scaling Laws (Hoffmann et al., 2022), i.e., $L(N, D) = E + \frac{A}{N^\alpha} + \frac{B}{D^\beta}$, where $N$ is the model size and $D$ is the number of training data. Under fixed batch sizes, we can treat the number of training data as the number of training steps $S$ as well. Notably, we do not directly fit the complete Chinchilla Scaling Law with two variables as we find it practically unstable to fit such many parameters simultaneously in our preliminary study, similar to the findings in Besiroglu et al. (2024). Instead, we decompose the law into two power laws for training steps $L(S) = E_1 + \frac{B}{S}$ and model sizes $L(N) = E_2 + \frac{A}{N}$, respectively. We first fit power laws of training steps to predict model performance with more training data then fit power laws of model sizes to predict the performance when scaling up models. We empirically find this routine stable.[9]

## 4.2 EXPERIMENT

We verify the effect of our pipeline with an experiment to minimize the validation loss of a 1B model trained on 100B tokens.

**Setups.** We train our models on the mixture of RedPajama and validate the validation set of the Pile to mimic the scenario where validation data are collected separately from the training data. To fit the scaling laws of training steps and model sizes, we train a series of 70M, 160M, 305M, and 410M models for 30B tokens. For all the models, we set the batch size as 1M tokens thus translating into

---

[8]An idea is to transfer the optimized training mixture on small models trained with few tokens to the training of large models and large volumes of training data. Nevertheless, as recent studies (Goyal et al., 2024; Kang et al., 2024; Covert et al.) highlight, the rankings of the data mixture vary as the model size and number of trained tokens change (Appendix C). Therefore, the optimal mixture at experimented scales can be suboptimal at the target scale.

[9]See Appendix D.2 for our preliminary verification. We notice some recent efforts try to investigate democratizing the implementation of predictions with scaling laws to facilitate applications (Su et al., 2024; Porian et al., 2024). While we illustrate our pipeline with the nested use of scaling laws, other implementations of scaling law predictions are also feasible and complementary to our method.

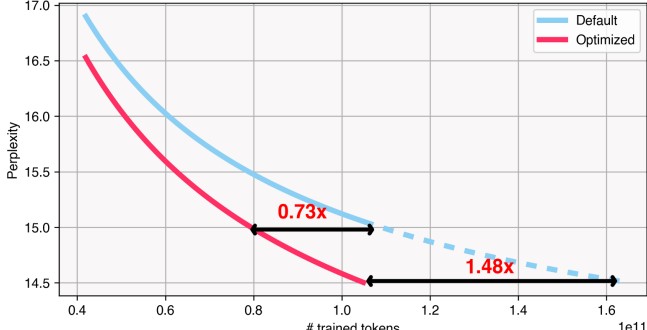

| Domains | Default Mixture | Optimized Mixture |
|---|---|---|
| CommonCrawl | 0.6700 | 0.1250 |
| C4 | 0.1500 | 0.2500 |
| Github | 0.0450 | 0.1406 |
| ArXiv | 0.0450 | 0.2500 |
| Books | 0.0450 | 0.0938 |
| StackExchange | 0.0250 | 0.1250 |
| Wikipedia | 0.0200 | 0.0156 |

Figure 8: The validation perplexity on the Pile validation set for 1B models trained on the default mixture and the optimized mixture of RedPajama for 100B tokens. Our optimized mixture achieves the performance of the default mixture only using 0.73 of the original number of training steps and eventually achieves a performance comparable to a default mixture trained with 1.48 times more tokens (estimated by the scaling law of training steps, shown as the dashed line). The specific mixture proportions are in the right table.

100k steps for the 1B models and 30k steps for small models. We apply a cosine learning rate decay with a warmup of 2k steps which decays to 0.1 of the maximum learning rate at the 100k-th steps.

To reach low prediction errors with a limited number of experiment runs, we select the mixtures for experimentation by leveraging the fact that mixture proportion terms are represented as exponential functions within our data mixing law. Specifically, we enumerate candidate mixtures by double-diminishing the proportion for each training domain, starting from the maximum available one that does not use up all the domain tokens. In this way, the losses of each (implicit) validation domain are distributed evenly. We then sample 40 mixtures from all the candidates and train the smallest 70M models. We resample groups of 20 mixtures from them to fit the data mixing law and select the group that reaches minimum prediction errors on all 40 samples as our final set of mixtures to run our pipeline. For more details, please refer to Appendix D.3.

**Results.** Our pipeline optimizes language modeling performance effectively. Fig. 8 shows the default mixture of RedPajama (Touvron et al., 2023a) in and the optimized mixture obtained from Alg. 1 with their performance on the validation data. The model trained on the optimized mixture can achieve a performance comparable to the one trained on the default mixture with only 73% steps. It eventually attains a performance that requires 48% more steps if trained using the default mixture. This indicates the effectiveness of our pipeline in mixture optimization[10].

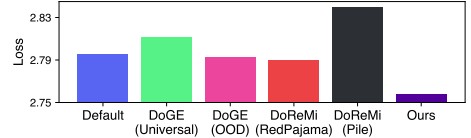

Figure 9: The language modeling performance of different data mixtures. All models are 1B models trained for 100B tokens with the same hyperparameters and validated on the validation set of the Pile. Specific proportions are in Fig. 21.

We also compare our optimized data mixture to previous optimization algorithms, which provide qualitative optimization. Specifically, we compare our method to DoGE (Fan et al., 2024) and DoReMi (Xie et al., 2024a). For DoGE, we compare both their universal generalization setting which assumes i.i.d. between training and validation data, and the OOD setting which optimizes for a given validation set, similar to ours. For DoReMi, which only works for universal optimization that ignores the validation data, we experiment on both a mixture optimized exactly on RedPajama and a mixture adapted from the one optimized on the Pile using the domain overlap between RedPajama and the Pile. More specific details on obtaining these data mixtures are in Appendix F.4. As shown in Fig. 9, our method finds the mixture that reaches the lowest losses for the same model sizes trained with the same data budgets. This further verifies the effectiveness of our method.

---

[10]The loss predictions are in Fig. 22, which shows the predictions are plausibly accurate and are consistent with the rankings of actual runs.

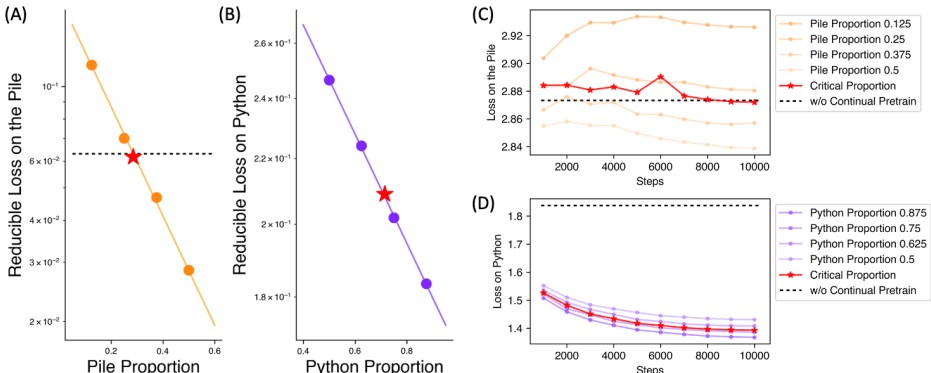

Figure 10: Loss predictions and the training curve of continual pretraining Pythia-70M on a mixture of the Pile and python code. (A) Loss prediction on the Pile; (B) Loss prediction on python; (C) training curves with losses on the Pile; (D) training curves with losses on python. We predict final losses with Eqn. 6. The law accurately finds the critical mixture proportion that maintains model performance on the original domain (i.e., the Pile).

## 5 DATA MIXING LAWS HELP AVOID CATASTROPHIC FORGETTING IN CONTINUAL PRETRAINING

We are also interested in applying our data mixing laws to continual pretraining, which shares the same paradigm as pertaining but begins the model with pretrained parameters instead of random initialization. Generally, continual pretraining is a common technique to enhance existing pretrained models. It injects up-to-date knowledge into the model, avoiding performance degradation due to distribution shifts (Gururangan et al., 2020; Xiong et al., 2023). In addition, researchers also apply continual pretraining to reuse existing model parameters to build models of a different architecture (Komatsuzaki et al., 2022).

We experiment on a typical scenario of continual pretraining, where we train the model on the mixture of original pretraining data and upcoming data of a target domain to enhance. For instance, we continually pretrain Pythia-70M models with a mixture of the Pile and Python codes, where the former is the original pretraining data of the base model. To verify whether our data mixing laws apply to continual pretraining, we train the models for 10B tokens on 4 mixtures and fit the Eqn. 6 on losses of the Pile and python codes. Results in Fig. 10 confirm that Eqn. 6 fits into the losses of continual pretraining.

During continual pretraining, a too-large proportion of the target data can hurt the performance of the original data. A representative mixture optimization target is to maintain the general-purpose ability (losses on the Pile) unchanged. To this end, using the fitted data mixing laws, we predict the *critical proportion* leading to the same loss as before continual pretraining. Fig. 10 demonstrates the success of our prediction where the proportion we find results in similar performance compared to the model before continual pretraining while gaining improvement in the target domain.

**Remarks.** We suggest continual pretraining is significant for its connection to the design of data schedules (Albalak et al., 2023; Chen et al., 2024b). Usually, continual pretraining applies to a pretrained model, while it is natural to further continually pretrain the continual pretrained models, i.e., multi-stage pretraining (Chen et al., 2024b). In each stage, the mixture proportions or even the domain components of training data can be different. This becomes a dynamic data schedule as the number of training stages approaches the infinite limit. Therefore, the successful application of our data mixing laws on continual training signifies a promising prospect for using it to design dynamic data schedules, a more comprehensive data curating paradigm.

## 6 DISCUSSIONS

In this work, we discover the quantitative predictability of model losses regarding the mixture proportions of training data, which boils down to the data mixing laws. Using data mixing laws allows practitioners to quantitatively estimate the model performance on unseen mixture proportions before the actual training, allowing low-cost tuning of data mixture together with scaling laws. Given the burgeoning interest in data engineering, we hope that our study paves the way for further quantitative inquiries and theoretical analyses in this research area.

ACKOWLEDGEMENT

This work was supported by the National Key Research and Development Program of China (No.2022ZD0160102). The computations in this research were performed using the CFFF platform of Fudan University. We thank Botian Jiang, Shiduo Zhang, and anonymous reviewers for their constructive feedback.

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

# A  RELATED WORK

**Curating pretraining data for LLMs.**  Training massive transformer architecture on trillions of tokens, a.k.a. pretraining, is the primary step to building modern large language models that exhibit impressive human-like generalist abilities (Brown et al., 2020; OpenAI, 2023; Jiang et al., 2023; Touvron et al., 2023b; Sun et al., 2024)). It takes up most of the computation resources for model training and researchers believe it endows almost all the knowledge in LLMs (Zhou et al., 2024). Such critical impact motivates the development of data curating strategies to reduce computation costs and enhance knowledge (Longpre et al., 2023). The efforts can be categorized into two steps. The first step focuses on obtaining a high-quality training dataset. A typical procedure includes selecting data sources to constitute different domains, deduplication, and the most intricate filtering (Wenzek et al., 2019; Penedo et al., 2023). A mass of endeavors in existing practice has involved multifarious filters, scoring the documents with from superficial features on characters (Rae et al., 2021; Xie et al., 2024b; Raffel et al., 2020) to semantics including similarity to the high-quality reference corpus (Wenzek et al., 2019) and toxicity (Longpre et al., 2023; Friedl, 2023). With a dataset on hold, the second step aims to make the best use of it. This includes tuning the data mixture (Du et al., 2022; Touvron et al., 2023a; Xie et al., 2024a) and devising data schedules (Mindermann et al., 2022; Albalak et al., 2023; Chen et al., 2024b; Fan et al., 2024). Our work is among those tune data mixtures and our extension to continue pretraining signifies our prospect of guiding the schedule design. Different from existing attempts that rely on intuition or qualitative targets, our study seeks a quantitative solution. As a concurrent work, Liu et al. (2024) also proposes to predict optimal data mixtures through regression but assuming rank invariance across training scales.

**Scaling laws**  are functional relationships between the properties of interests (e.g., test loss or other performance metrics) and the scales of controllable factors regarding the optimization process or architecture (e.g., model sizes and numbers of training samples) (Villalobos, 2023). Along with the development of machine learning, characterizing scaling behaviors has garnered great research interest under the context of learning theories, bounding the generalization error given the number of training samples in the form of power laws (Vapnik & Chervonenkis, 1971; Valiant, 1984; Haussler, 1988; Amari et al., 1992). Nevertheless, overly strict assumptions hinder their practical applications. In recent years, statistical estimation on scaling gained fast progress for deep neural networks and spawns the introduction of scaling laws. Hestness et al. (2017) pioneers the trend and demonstrates power-law generalization error scaling across a breadth of factors but the power-law exponents differ from previous theoretical analysis. Kaplan et al. (2020); Hoffmann et al. (2022); Henighan et al. (2020) conduct more comprehensive investigations on Transformer architecture (Vaswani et al., 2017), further highlighting the power-law relationship on test loss regarding model sizes, the amount of training data and computation across orders of magnitudes. These findings foretell the performance gain with scaling quantitatively and guide the trade-off between larger models and more training data, directing to the later development of large language models (Brown et al., 2020; Hoffmann et al., 2022; OpenAI, 2023). Lately, progressive investigations propose amendments to existing scaling laws (Caballero et al., 2022; Alabdulmohsin et al., 2022), seeking theoretical explanations on the empirical formulas Bahri et al. (2021); Hutter (2021); Michaud et al. (2024), and exploring the functional relationships in broader scenarios (Hernandez et al., 2021; Frantar et al., 2023; Liu et al., 2023). The most relevant study to ours is Hashimoto (2021) which explores performance prediction under multiple data sources but is limited to small-scaled supervised learning tasks.

# B  LIMITATIONS AND DISCUSSIONS

How data mixtures affect model training is far from fully understood. Our study makes preliminary attempts at a quantitative framework while leaving several limitations.

**On the clarification of domains.**  The concept of domains is not well-defined. In this paper, similar to related studies (Xie et al., 2024a; Chen et al., 2024b; Albalak et al., 2023; Fan et al., 2024), we directly adopt the predefined domains in the open-source training data. Nevertheless, we suppose that more operationally defined training domains, e.g., clustering (Gururangan et al., 2023; Shao et al., 2024), could further benefit the performance of outcome models. For the validation domains, our implicit domain aggregation method obviates the necessity of explicitly aligning validation data with training domains. This requirement is often encountered, given that validation data typically comprises trustworthy datasets rather than mere compilations from training domains. However, we

acknowledge that implicit domain aggregation may be less interpretable compared to the explicit approach and may raise concerns regarding its accuracy, as elaborated subsequently.

**On the error analyses.** Leveraging scaling laws requires experiments to provide samples to fit the functions. Consequently, it requires careful design of experiments (Mead, 1990) to decide the number of fitting samples to experiment with and how to distribute these samples to reduce prediction errors to the greatest extent. In this study, we decide the number according to our affordable budget and leverage the simple rule that evenly distributes the losses of the data samples but considering more theoretically justified rules should be necessary. Additionally, our nested use of scaling laws can introduce errors in each step. Therefore, further analyses to mitigate the error accumulation are also demanding. In Fig. 22, we notice our predictions are smaller than the actual loss, which we attribute to the underestimation from the step laws and model size laws we fit. Further practical experience demystifies the technical details of scaling laws (Su et al., 2024) can help eliminate the errors.

**On joint laws of multiple factors.** We propose the nested use of scaling laws for circumventing the difficulties in finding a joint law of training steps, model sizes, and mixture proportions. Although we can predict the losses with our pipeline, a joint law unveils clear synergies of different factors. For instance, previous studies indicate the power-law exponent in the scaling laws of model sizes and training data are insensitive to training and validation data (Hestness et al., 2017; Kaplan et al., 2020; Hashimoto, 2021; Hoffmann et al., 2022; Frantar et al., 2023). Figuring out their joint laws with data mixture can further confirm this surmise. Moreover, a joint law also implements coefficient-sharing of separate laws, reducing the number of required fitting samples.

**On dynamic data curating.** Our study presents a pipeline to decide on a group of fixed mixture proportions for pretraining. More sophisticated data curating can include dynamic proportions (Albalak et al., 2023) and even a curriculum that changes data domains (Chen et al., 2024b). The application of our data mixing laws in continual pretraining (Sec. 5) implies the prospect of extending our findings to these settings. On top of this, we believe that it is promising to incorporate further analysis to pursue a dynamic data mixing law.

**On theoretical understandings.** Our data mixing laws, similar to most scaling laws, are empirical findings. We believe a theoretical understanding of the training dynamics that form the laws provides a more solid justification. A potential perspective is understanding the target of tuning mixture proportion through gradient estimation (Guo et al., 2024; Gu et al., 2024). Specifically, the mixture proportions weight data from different domains, whose effect boils down to the weight for the linear combination of gradients from different domains during training. This perspective turns the target of tuning mixture proportions into finding an ideal gradient direction (Gu et al., 2024) and the relationship between data samples is formalized with their gradient directions (Guo et al., 2024).

We believe that further investigation into these issues could facilitate more sophisticated quantitative methods for data engineering. We leave them as future work.

## C   THE RANKING OF DATA MIXTURES DEPEND ON MODEL SIZES AND TRAINING STEPS.

One may wonder whether we can find the optimal data mixtures on small models and few numbers of steps, and then transfer the found mixture proportions to large-scale training. To answer this question, we compare the relative performance of models in different sizes and trained with different numbers of steps in Fig. 11.

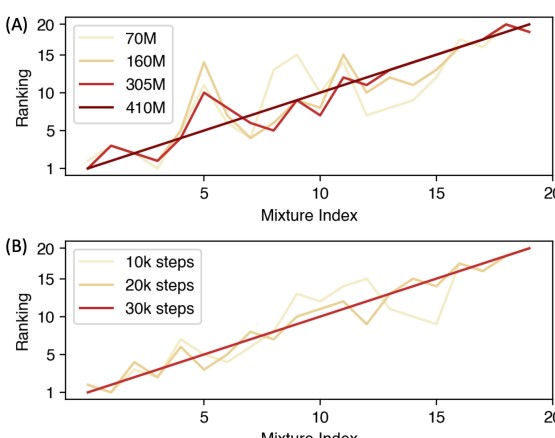

Results show that the relative performance fluctuates despite a relatively consistent trend across sizes and training steps. This indicates that a mixture is better at small scales but does not always perform better at large scales, consistent with findings of Goyal et al. (2024); Covert et al.; Kang et al. (2024). The longest common sequence of the partial orders among the 20 mixtures in Fig. 11(A) and Fig. 11(B) only reaches lengths of 10 and 11, respectively.

Figure 11: The rankings of the relative performance of 20 sample mixtures trained on RedPajama and validate on the Pile. **(A)** The rankings of models of different sizes all trained for 30k steps. **(B)** The rankings for 70M models trained for different steps.

## D   IMPLEMENTATION DETAILS

### D.1   MODEL TRAINING

Throughout this study, we employ the Pythia Suit (Biderman et al., 2023) as our model architectures, the specific configurations are in Tab. 2. The maximum sequence length is 4096 for pretraining from scratch and 2048 for continual pretraining, where the latter aligns with the setting of the original pretrained models. In all our experiments, we train the model with a batch size of 1M tokens and a maximum learning rate of 1e-4. We warm up the learning rates for 2000 steps and decay it to 0.1 of the maximum at the last training step with a cosine decay schedule. For continual pretraining, we initialize the models with the 20k-step checkpoint of the Pythia 70M model and do not apply a learning rate warmup. For the costs of our experiments, it takes around 3.5/8/16/21 hours to train a 70M/160M/305M/410M model for 30B tokens on 8 A100 GPUs on our infrastructure.

Table 2: Model architectures for experiments in this paper.

|                      | 70M        | 160M       | 305M        | 410M        | 1B          |
| -------------------- | ---------- | ---------- | ----------- | ----------- | ----------- |
| Vocabulary Size      | 50304      | 50304      | 50304       | 50304       | 50304       |
| Non-embedding Params | 18,915,328 | 85,056,000 | 201,541,632 | 302,311,424 | 805,736,448 |
| Layers               | 6          | 12         | 16          | 24          | 16          |
| Model Dimension      | 512        | 768        | 1024        | 1024        | 2048        |
| Heads                | 8          | 12         | 16          | 16          | 8           |

For datasets, we mainly experiment with the Pile and RedPajama. For the Pile, we find duplicates in the raw data, so deduplication is performed before training with it. The Pile contains 5 coarse-grained domains, which are further decomposed into 22 fine-grained domains. Our experiment in Sec. 3.1 is on Github and Pile-CC domains while the experiment in Sec. 3.2 is on Github, Pile-CC, and the Books. All these are fine-grained domains. For our experiments with 5 domains in Sec. 3.3 we adopt the five coarse-grained domains, i.e., academic, internet, prose, dialogues, and misc, where misc include Github and the DeepMind Mathematics Dataset which are symbolic content. We use the coarse-grained domains because it is hard to find five fine-grained domains with sufficient tokens. For the RedPajama, we download the version produced and shared by Chen et al. (2024a).

## D.2 PREDICTING LANGUAGE MODELING PERFORMANCE WITH SCALING LAWS

In our prediction pipeline in Sec. 4, we adopt nested use scaling laws of training steps and model sizes, which are both power laws, to predict language modeling performance at scale. To fit the laws, we follow Hoffmann et al. (2022) to search over a set of initialized parameters and fit the samples by minimizing the Huber errors between predictions and observations with LBFGS.

We present our results on verifying the feasibility of applying scaling laws to predict language modeling performance. Our prediction pipeline (described in Sec. 4) employs two scaling laws: one related to training steps and another to model sizes, to extrapolate performance with increased training data and larger models. We evaluate the precision of predictions for each of these scaling laws, respectively.

**Scaling laws of training steps.** Fig. 12 shows the training curve 70M models on three different data mixtures. We fit power laws within 30k steps (marked with circles) and extrapolate to predict model performance to as many as 100k steps (marked with stars). On all validation sets, the fitted curves give descent prediction precision, with a low mean absolute error of 0.02.

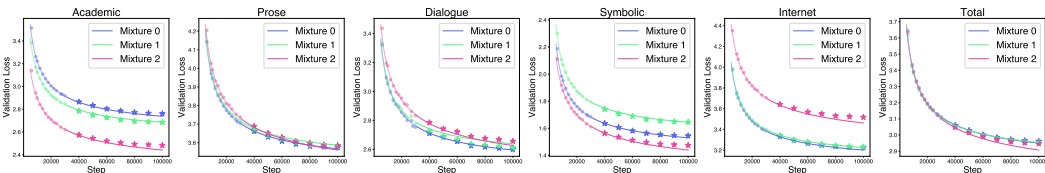

Figure 12: Verification on predicting language modeling performance with scaling laws of training steps. We train 70M models on three different mixtures up to 100k steps and validate them on 5 validation domains as well as the overall validation mixture. All curves are fitted within 30k steps (•) and and extrapolated to predict model performance to 100k steps (⋆)

**Scaling laws of model sizes.** Fig. 13 shows the results where we fit power laws on 70M, 160M, and 305M models (marked with circles) and extrapolate the curve to predict 410M model performance (marked with stars) at different training steps and under different data mixtures. The results are positive, showing that we can precisely predict the 410M model performance at different training steps, with a mean absolute error of 0.003.

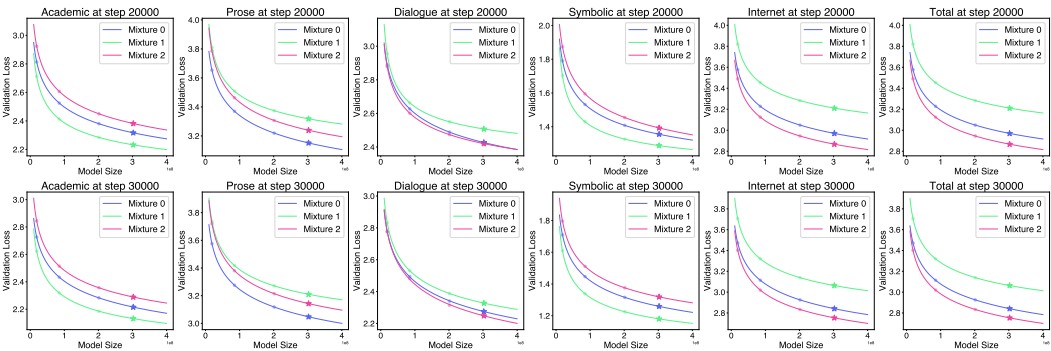

Figure 13: Verification on predicting language modeling performance with scaling laws of model sizes. We train models of 70M, 160M, and 406M on three different mixtures and validate them on 5 validation domains as well as the overall validation mixture. All curves are fitted with models of 70M, 160M, and 305M (•) and extrapolated to predict the performance of 410M models (⋆). We verify the predictability at different numbers of training steps.

Overall, we consider fitting power laws to predict model performance for more training steps and larger models are feasible. Therefore we adopt them to implement the scaling law predictions in our pipeline (Sec. 4).

---

**Algorithm 2** Sampling mixture proportions for fitting mixture laws.

---

**Input:** Maximum proportions of $M$ domains $\boldsymbol{r}_{max} = [r_{max,1}, \ldots, r_{max,M}]$, where $r_{max,i} = \frac{D_i}{D_{target}}$ with $D_i$ and $D_{target}$ being numbers of available tokens in $i$-th domain and target number of training tokens, respectively, sorted in descending orders (i.e., $r_{max,1} \geq r_{max,2} \geq \cdots \geq r_{max,M}$), minimum proportion grid size $\delta$, number of mixture to run experiment $N$.

**Output:** A set of $N$ mixtures to experiment $\{\boldsymbol{r}_n\}_{n=1}^N$.

1: Candidate mixtures $\mathcal{C} \leftarrow$ GETALLCANDIDATES$(1, [])$
2: Split mixtures with 0 proportion in $\mathcal{C}$ into $\mathcal{C}_0$ and the others into $\mathcal{C}_1$
3: Samples $\{\boldsymbol{r}_n\}_{n=1}^{\lfloor N/4 \rfloor}$ from $\mathcal{C}_0$ and $\{\boldsymbol{r}_n\}_{n=\lceil N/4 \rceil}^N$ from $\mathcal{C}_1$
4:
5: **procedure** GETALLCANDIDATES(domain index $i$, proportions of first $i-1$ domains $r_{1\ldots i-1}$)
6:     Candidate mixtures $\mathcal{C} = \emptyset$
7:     **if** $i = M$ **then**
8:         **if** $0 \leq 1 - \sum_{j=1}^{i-1} r_j \leq r_{max,i}$ **then**
9:             $r_{1\ldots i} \leftarrow [r_{1\ldots i-1}, 1 - \sum_{j=1}^{i-1} r_j]$
10:             $\mathcal{C} \leftarrow \mathcal{C} \bigcup \{r_{1\ldots i}\}$
11:         **end if**
12:     **else**
13:         $\Gamma \leftarrow \delta * \lfloor \frac{r_{max,i}}{\delta} \rfloor$
14:         **for** $s = 0$ **To** $\lceil \log_2 \frac{\Gamma}{\delta} \rceil$ **do**
15:             $r_i \leftarrow \max(0, \frac{\Gamma}{2^s})$
16:             $\mathcal{C} \leftarrow \mathcal{C} \bigcup$ GETALLCANDIDATES$(i+1, [r_{1\ldots i}])$
17:         **end for**
18:     **end if**
19:     **return** $\mathcal{C}$
20: **end procedure**

---

### D.3 FITTING DATA MIXING LAWS

Fitting the mixture law requires us to first experiment on a few mixtures and obtain their losses. The sample mixture chosen for fitting could largely affect the prediction accuracy. Consider an extreme case where all sample mixtures have proportions around a small region, it is hardly possible to fit a law that reliably predicts the whole proportion space.

In this paper, we intuitively try evenly allocating the mixture proportions regarding their losses. Specifically, we enumerate candidate mixtures by double-diminishing the proportion of each domain so that the losses are distributed evenly among these mixtures. Then, according to the available computation budget, we sample a certain number of mixtures from the candidates to run experiments. During sampling, we find candidate mixtures with a 0 domain proportion in any of the training domains take up a majority of the candidates. To avoid these candidates making up all our samples, we specifically down-sample them. The concrete algorithms are in Alg. 2. Additionally, we employ AdaBoost Regressor (Drucker, 1997) for fitting the mixture laws to stabilize the predictions and improve their accuracy. We encourage future studies to dive into a more careful design of candidate mixture selection with theoretical support.

## E CONNECTIONS BETWEEN IMPLICIT DOMAIN AGGREGATION AND MLP

We first repeat our final mixture law (Eqn. 8) here for convenience:

$$L(r_{1\ldots M}) = \sum_{i=1}^K s_i L_i(r_{1\ldots M}) = \sum_{i=1}^K s_i \left[ c_i + k_i \exp\left(\sum_{j=1}^M t_{ij} r_j\right) \right],$$

where $r_{1\ldots M}$ are mixture proportions on $M$ training domains, $L_i$ are validation loss on $K$ implicit domains with $s_i$ as their weight in the overall validation set, and $c_i, t_{ij}$ are other parameters to fit.

The mixture law boils down to a computation graph in Fig. 14, which contains two layers. The first layers predict the domain losses, while the second sums up the domain losses to obtain the overall validation loss. In this way, the mixture law becomes a multilayer perception (MLP) with an exponential activation function. In practice, we fit the mixture laws with implicit

domain aggregation by fitting a multilayer perception with exponential activation and applying softmax to the output layer weights. Additionally, considering the high variance of MLP, we further employ AdaBoost Regressor (Drucker, 1997) for fitting the mixture laws to stabilize the predictions and improve their accuracy.

Inspired by this perspective, we attribute the successful fitting of data mixing laws to two aspects. First, the MLP with a sufficiently large hidden dimension is a universal approximator (Hornik et al., 1989) thus being able to fit the relationships between losses and mixture proportions. Second, the mixture proportions are bounded between 0 and 1. For this reason, predicting an unseen mixture is an interpolation problem, which is usually easier than extrapolation.

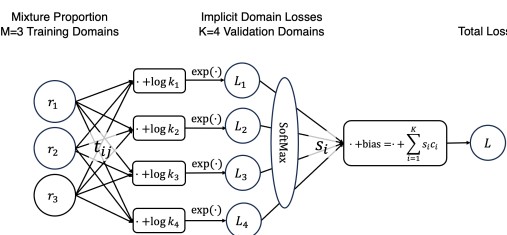

Figure 14: The computation graph of mixture law with implicit domain aggregation. We take an case of 3 training domains and 4 implicit validation domains as example. The parameters correspond to the notations in Eqn. 8.

# F  SUPPLEMENTED RESULTS

## F.1  PREDICTION RESULTS ON MORE DOMAINS

To further consolidate the efficacy of data mixing laws and show that they are general for different data, we experiments on domains different from those in Sec. 3.2.

We train and validate on Wikipedia, ArXiv, and StackExchange of RedPajama, which are three domains different from those in Sec. 3.2. All samples are from 70M models trained for 10k steps. The prediction accuracy is in Fig. 15. The result shows the predicted and observed losses are consistent for different mixtures. This confirms that our data mixing laws also work on domains besides those in the main experiments.

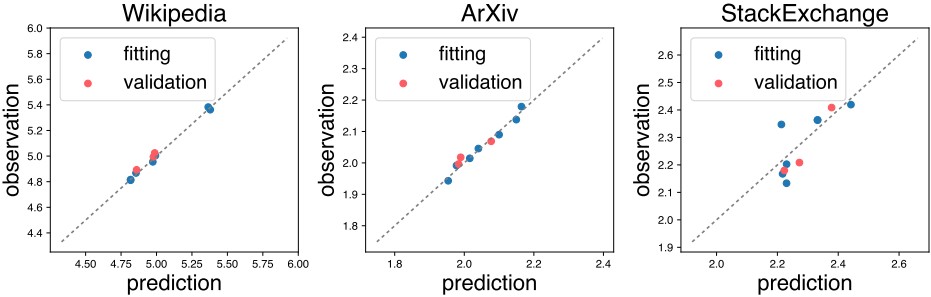

Figure 15:  Prediction results on domain losses with Eqn. 7. We train 70M models on mixtures of Wikipedia, ArXiv, and StackExchange for 10k steps. We fit on 7 mixtures and validate on 3 other mixtures.

## F.2  DATA MIXING LAWS MAKE NO DOMAIN-INDEPENDENT ASSUMPTIONS

Although our data mixing laws combine the terms with the proportion of different domains through linear combination, we make no domain-independent assumption that different domain affects the losses independently. This is because the linear combination serves as an exponent in Eqn. 6 and Eqn. 7. Specifically, by Taylor expansion, we have

$$L_i(r_{1...M}) = c_i + k_i \exp\left(\sum_{j=1}^{M} t_{ij}r_j\right) = c_i + k_i\left(1 + \sum_{j=1}^{M} t_{ij}r_j + \frac{1}{2}\sum_{j=1}^{M}\sum_{k=1}^{M} t_{ij}t_{ik}r_jr_k + o^2\right),$$

where there exists interaction terms $r_jr_k(j \neq k)$ of different mixture proportions.

Empirically, we evaluate the effectiveness of our data mixing laws in modeling domain interactions by examining their ability to predict language modeling performance when mixing two correlated domains. Specifically, we construct two synthetic data domains with deliberate overlap. The first domain consists of 50% Wikipedia and 50% CommonCrawl data, while the other domain comprises 50% Wikipedia and 50% ArXiv content. In this case, increasing the proportion of one domain necessarily increases the shared Wikipedia component. Therefore, the contribution of a training domain on target losses is coupled with the proportion of the other domain given their joint contribution through Wikipedia. As demonstrated in Fig.16, our proposed mixing law (Eqn.6) successfully models the language modeling performance across various mixing ratios of these correlated domains.

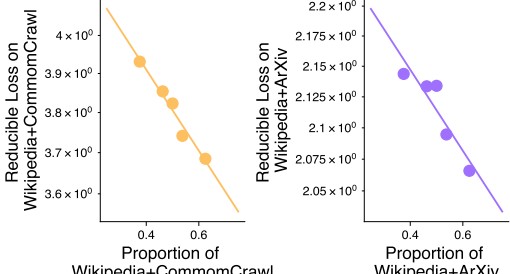

Figure 16: Data mixing laws can model the language modeling performance of mixing correlated domains with different proportions. We train 70M models on the mixtures of "Wikipedia+ Common-Crawl" and "Wikipedia+ArXiv" for 15k steps. We validate on the two domains separately and fit the relationship between mixture proportions and validation losses with Eqn. 6.

## F.3  EXTRA VALIDATION ON SCALING LAWS PREDICTION

We discuss the computation that our prediction method with nested scaling laws requires. In particular, the cost primarily depends on how much scaling laws can accurately extrapolate.

Specifically, we need to train $N$ different data mixtures on model sized $N_1, N_2, \ldots, N_K$ for $S_0$ steps to predict the model performance of different data mixtures trained with a model with $N_{target}$ parameters for $S_{target}$ steps. The total extra computational overhead relative to direct training is $N \frac{S_0 \sum_{i=1}^{K} N_i}{S_{target} N_{target}}$, where the fraction $\frac{S_0 \sum_{i=1}^{K} N_i}{S_{target} N_{target}}$ represents computation saved through scaling law predictions. State-of-the-art scaling law prediction demonstrates that this fraction can be 1/100 to 1/1000 (OpenAI, 2023; Bi et al., 2024). Together with the typical value of $N$, which is 20 in our experiments, the overall method should require an extra 1/5 to 1/50 training computation expectedly.

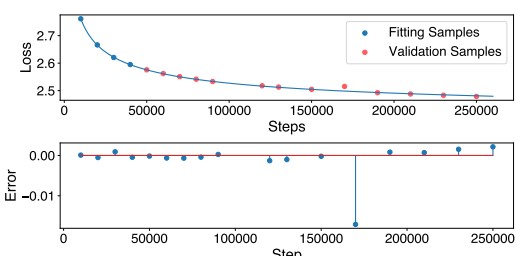

Figure 17: The scaling law of training steps accurately extrapolates to 6.25x more steps. We fit $L(S) = E + B/S^\beta$ with 40k training steps of a 1B model and validate the prediction on language modeling performance up to 250k steps.

Given that achieving accurate scaling law predictions remains a developing area, we would like to provide our preliminary investigation to support 100x to 1000x scaling. Fig. 17 shows the scaling prediction of training steps with the scaling law of training steps $L(S)$, where we fit with the first 40k steps and predict the model performance up to 250k steps. This shows that fitting with 40k steps accurately predicts the language modeling performance on 250k steps, which is 6.25x scaling. Additionally, Fig. 18 shows the scaling prediction of model sizes with L(N), where we fit with models smaller than 100M and find it accurately predicts model performance up to 7.25B, which is 72.5x scaling. Combining L(S) and L(N), we may achieve 450x scaling.

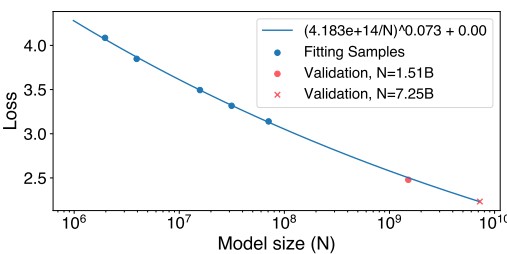

Figure 18: The scaling law of model sizes accurately extrapolates to 70x larger models. We fit language modeling performance at convergence following (Kaplan et al., 2020) with $L(N) = B/N^\alpha + E$. The language modeling performance of 1.5B and 7.25B models are predicted with L(S).

### F.4 COMPARISON TO OTHER DATA MIXING METHODS

We compare our method to representative data mixing methods, DoGE (Fan et al., 2024) and DoReMi (Xie et al., 2024a). As our experiment in Sec. 4.2, we train on RedPajama and validation on the Pile.

DoGE (Fan et al., 2024) contains a universal generalization setting, which assumes validating on the same data as training, and an OOD setting which targets any validation data. We experiment with both of them. For universal generalization, we refer to the data mixture provided by Fan et al. (2024). For the OOD setting, we follow the original paper to train a small proxy model (160M) for 10k steps and apply their online updating rule to adjust the data mixture, shown in Fig. 19. We also follow Fan et al. (2024) to calculate the average proportions along the training steps of the proxy model as the final optimized mixture.

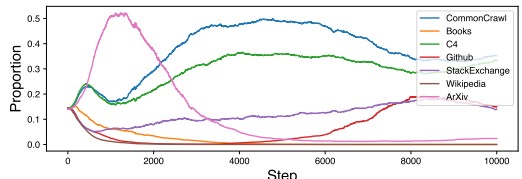

Figure 19: The evolution of mixture proportions when training the proxy model with the updating rule in the OOD setting of DoGE.

For DoReMi (Xie et al., 2024a), which is only designed for general optimization without awareness of the validation data, we experiment on both its mixture proportion optimized with RedPajama and the Pile. For the mixture optimized with RedPajama, we adopt the result of DoReMi10k from Fan et al.

(2024). For the mixture optimized on the Pile, we refer to the optimized Pile mixture in the original paper (Xie et al., 2024a) and adapt the mixture to the one for RedPajama according to the domain overlap. Specifically, for ArXiv, Wikipedia, Github, and StackExchange, we directly borrow the mixture proportion. CommonCrawl and C4 equally share the proportion of Pile-CC. The proportion of Books is obtained as the sum of Books3 and BookCorpus2 in the Pile. We renormalize the proportions of these domains to ensure they sum up to 1.

Fig. 21 summarizes the final mixture proportion we use for different setups. We train all models for 100B tokens at the model size 1B. The outcome performance is in Fig. 20 which shows the mixture provided by our data mixing law indeed archives the lowest validation loss.

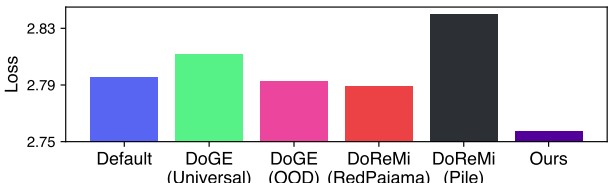

Figure 20: Comparisons of the language modeling performance of different data mixtures. All models are 1B models trained for 100B tokens with the same hyperparameters and validated on the validation set of the Pile. **Default**: original data mixture from Touvron et al. (2023a). **DoGE (Universal)**: DoGE for universal generalization, obtained from Fan et al. (2024). **DoGE (OOD)**: OOD generalization setting of DoGE optimized with validation set of the Pile. **DoReMi (RedPajama)**: DoReMi mixture optimized by training proxy model on RedPajama. **DoReMi (Pile)**: DoReMi mixture optimized by training proxy model on the Pile and adapted for our training on RedPajama through the domain overlaps between two dataset. Specific proportions are in Fig. 21.

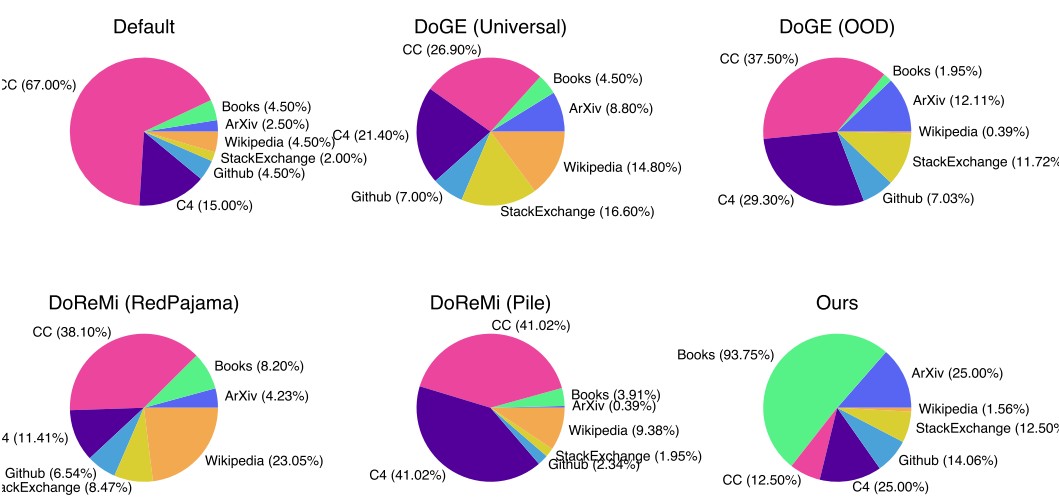

Figure 21: Specific mixture proportions on Redpajama from different data mixture optimization methods.

# G    LOSS PREDICTION RESULTS WITH NESTED SCALING LAWS

Fig. 22 shows the prediction results of nested use of scaling laws in Sec. 4.2. The result demonstrates plausible reference on the relative scale of losses on both the overall validation data and different validation domains. The optimized mixtures perform better in most domains. While the overall loss helps optimize the overall performance, losses on different domains show model capabilities in various aspects. Our result indicates that, by tuning data mixtures, it is possible to improve specific model capabilities without sacrificing others, consistent with the findings of Xie et al. (2024a).

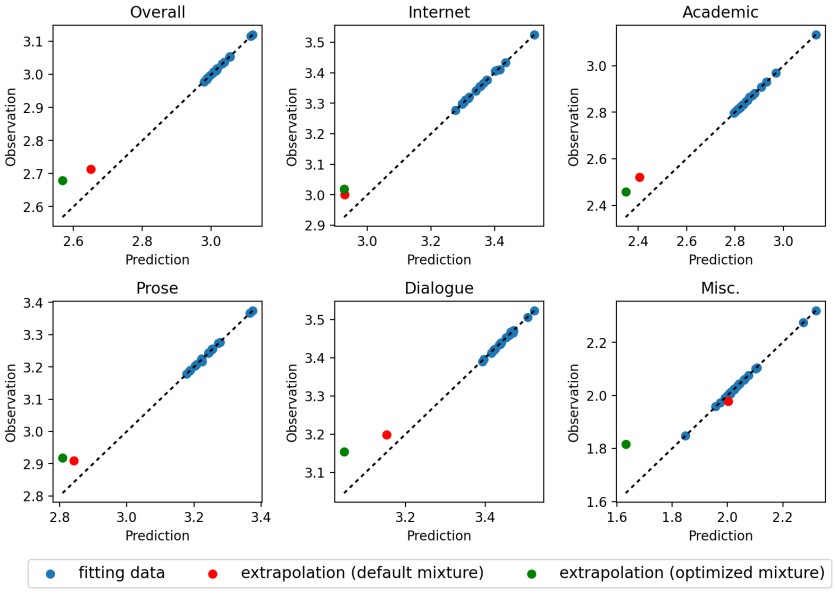

Figure 22: Results of our loss prediction pipelines for the overall validation loss and domain losses. Fitting data are from 70M to 410M models trained for 30B tokens, while the extrapolated points are from the default and optimized mixture for 1B models and 100B tokens.

