# OpenReview forum: "Data Mixing Laws: Optimizing Data Mixtures by Predicting Language Modeling Performance"
_ICLR.cc/2025/Conference — ICLR 2025 Poster_

### Official Review · Reviewer_dZPn · 2024-10-28

**Soundness:** 2
**Presentation:** 3
**Contribution:** 2
**Rating:** 6
**Confidence:** 4

**Summary:**

The paper extends the scaling law approach to optimize data mixtures for efficient language model pretraining. Specifically, the authors fit a scaling law function (i.e. data mixing laws) between the validation loss and training in-domain sampling ratios. From extensive experiments, the approach can improve language model pretraining efficiency on RedPajama dataset, as well as mitigating the catastrophic forgetting by training on mixture of generic dataset and domain specific data.

**Strengths:**

1. The paper is well-written, where the methodology and experimental results are clearly demonstrated.
2. The effectiveness from the method is well-justified from extensive experiments on LLM pretraining and continual pretraining.

**Weaknesses:**

1. **Lack of a detailed analysis on computation overheads**: how many small-scale experiments are required to get an accurate prediction of the scaling law function? If the default baseline train with these extra computation flops, how would the performance comparing with the proposed method?

2. **unfair comparison between some baselines**: the DOREMI [1] and DOGE [2] weights from Fan et al.'s paper are both targeting at minimizing the model's performance on SlimPajama dataset, which has a great domain shift to the target used by the authors (the Pile). I would suggest to use the DOREMI weights from their original paper [1] which also optimized on the Pile and retrain DOGE with target domain as your validation set from the Pile.

3. trivial points:
    - the footnotes (9,10) are not properly displayed on the paper.

[1] DoReMi: Optimizing Data Mixtures Speeds Up Language Model Pretraining
[2] DoGE: Domain Reweighting with Generalization Estimation

**Questions:**

1. **Computation costs analysis**: how many small-scale experiments are required to get an accurate prediction of the scaling law function? If the default baseline train with these extra computation flops, how would the performance comparing with the proposed method?
2. **Fair comparison between DOREMI and DOGE**: if apply the doremi weights from original doremi paper and doge weights using the correct validation domain, how the performance of the proposed data mixing law comparing to them?

---

> ### Author Response · Authors · 2024-11-19
> **Responses to Reviewer dZPn (1/2)**
>
> Dear Reviewer dZPn,
>
> Thank you for your effort in reviewing our paper and providing valuable suggestions. We summarize your questions and answer them as follows.
>
> > Q1: Lack of a detailed analysis on computation overheads
>
> Thank you for your suggestion on analyzing the computation overheads. **We include detailed experiments and discussion in Appendix F.3 of our updated manuscript.**
>
> The extra computation required in our method is the number of fitting mixture multiplied by the costs to fit scaling laws for each mixture. We aim to optimize the data mixture of a $N_{target}$-sized model trained for $S_{target}$ steps, whose training computation can be estimated as $6N_{target}S_{target} B$ [1], where $B$ is the batch size.
> To predict the final results, we train small models of scale $\\{N\_i\\}\_\{i=1\}^K $ on N mixtures for $S_0$ steps. Which involves training computation $6BNS_0\sum_{i=1}^K{N_i}$. Therefore, the additional compuataion overhead would be $\frac{6BNS_0\sum_{i=1}^K{N_i}}{6N_{target}S_{target} B} = N\frac{\sum_{i=1}^K N_i}{N_{target}}\frac{S_0}{S_{target}}$, where  $N$ is the multiple introduced by data mixing laws, typically 20 in our experiments, while $\frac{\sum_{i=1}^K N_i}{N_{target}}\frac{S_0}{S_{target}}$ is the computation saved with scaling laws. With this fraction, we see that **the compuation budget of our method mostly depends on the extrapolation ability of scaling laws**.
>
> Regarding the concrete numbers in practice, state-of-the-art scaling predictions enable 100x to 1000x extrapolation [2,3]. This makes the full pipeline only involve 1/5 to 1/50 extra computation, which is much smaller than the extra compute for the default mixture to match our optimized mixture (i.e., 48%). Admittedly, we do not have an excellent scaling law prediction technique at the time we finish this study, which costs as much as 1/10 of the final computation for each data mixture. Nonetheless, our latest experiment also confirms that L(S) is able to achieve at least 6.25x scaling and L(N) 72.5x scaling, with results we update in **Appendix F.3**. We plan to update these analyses with more thorough study in later versions.
>
> Note that scaling prediction remains a developing research area[4,5,6], the accuracy and efficiency of our method can be further improved with more investigations into the practice of scaling-law prediction, which would also be a focus of our future work.
>
> [1] Scaling Laws for Neural Language Models
>
> [2] GPT4 Technical Report
>
> [3] DeepSeek LLM Scaling Open-Source Language Models with Longtermism
>
> [4] Unraveling the Mystery of Scaling Laws: Part I
>
> [5] Scaling Laws with Learning Rate Annealing
>
> [6] A Hitchhiker’s Guide to Scaling Law Estimation

---

> > ### Author Response · Authors · 2024-11-19
> > **Responses to Reviewer dZPn (2/2)**
> >
> > ---
> >
> > > Q2:  I would suggest using the DoReMi weights from their original paper which also optimized on the Pile and retrain DoGE with target domain as your validation set from the Pile.
> >
> > Thank you for your suggestions to help make comparisons stricter. **We follow your advice and update our results as follows, with details and visualization included in Appendix F.4 of our updated manuscript.**
> >
> > For DoGE, we follow train a small proxy model (160M) for 10k steps with their OOD version algorithm and obtain the data mixture as follows. We include the optimization process in Fig. 19.
> >
> >
> > |Domain|CC|Books|C4|GitHub|StackExchange|Wikipedia|ArXiv|
> > |-|-|-|-|-|-|-|-|
> > |Proportion|0.375|0.1953125|0.29296875|0.0703125|0.1171875|0.00390625|0.12109375|
> >
> >
> > For DoReMi, we leverage the domain overlap between the Pile and RedPajama datasets to adapt the optimized mixture on the Pile for RedPajama. Specifically, we borrow the proportion of GitHub, StackExchange, Wikipedia, and ArXiv. For CommonCrawl and C4, we let them share the proportion of Pile-CC. We also assign the proportion of Books as the sum of BookCorpus2 and Books3. With the adapted proportions, we normalized them to ensure they sum up to 1 and train with RedPajama with the normalized mixture proportions. The specific data mixture is as follows.
> >
> >
> > |Domain|CC|Books|C4|GitHub|StackExchange|Wikipedia|ArXiv|
> > |-|-|-|-|-|-|-|-|
> > |Proportion (before normalization)|0.6057 / 2 (half of Pile-CC)|0.0061+0.0224 (Books3+BookCorpus2)|0.6057/2 (half of Pile-CC)|0.0179|0.0153|0.0699|0.0036|
> > |Proportion (normalized)|0.41015625|0.0390625|0.41015625|0.0234375|0.01953125|0.09375|0.00390625|
> >
> > The training results are as follows, which confirms that **the data mixture given by our method achieves the lowest validation loss**.
> >
> > $~$
> >
> > Tab. 1 Comparison of different data mixture optimization methods. Default: the mixture originally used by LLaMa. DoGE (Universal): DoGE for universal generalization, which optimizes on a validation set i.i.d. to training data, we obtain the mixture from [1]. DoGE (OOD): DoGE optimized for a given validation data. DoReMi (RedPajama): DoReMi optimized for RedPajama. DoReMi (Pile): DoReMi mixture adapted from the results of optimizing on the Pile.
> >
> > | |Default|DoGE (Universal)|DoGE (OOD)|DoReMi (RedPajama)|DoReMi (Pile)|Ours|
> > |-|-|-|-|-|-|-|
> > |Validation Loss|2.795509|2.811973|2.792664|2.789486|2.840071|**2.757676**|
> >
> > [1] DoGE : Domain Reweighting with Generalization Estimation
> > [2] DoReMi: Optimizing Data Mixtures Speeds Up Language Model Pretraining
> >
> > ---
> > We hope our supplemented discussions help address your concern. We are more than delighted to discuss with you and address further concerns, if any.  Looking forward to your feedback.

---

> > > ### Comment · Reviewer_dZPn · 2024-11-20
> > >
> > > Thanks the author for the clarification of computational overheads and additional experiment results, which mostly resolve my concerns. While the extra computations is still intensive comparing to proxy-based baselines (e.g. DoReMi and DoGE), it is a common pitfall for all scaling-law based method.
> > >
> > > I am happy to raise my score from 5 to 6.

---

> > > > ### Author Response · Authors · 2024-11-21
> > > >
> > > > We are glad that our discussion and the additional experimental results can resolve your concerns.
> > > >
> > > > Thank you so much for taking the time to carefully review our manuscript and for your constructive feedback. And we remain committed to further improving the work based on your valuable insights.

---

### Official Review · Reviewer_uTnD · 2024-10-30

**Soundness:** 3
**Presentation:** 3
**Contribution:** 3
**Rating:** 6
**Confidence:** 4

**Summary:**

This paper proposes a novel framework for optimizing data mixtures in pretraining large language models (LLMs). The authors introduce data mixing laws—predictive functions that estimate model performance based on the proportions of different data domains, allowing for the prediction of model outcomes across unseen data mixtures without extensive re-training.

Key contributions include:

1. Data Mixing Laws: The authors formalize a quantitative relationship between data mixture proportions and model losses, termed "data mixing laws." These functions enable the prediction of model performance with different data mixtures.

2. Pipeline for Scalable Predictions: By leveraging nested scaling laws for training steps, model sizes, and data mixtures, the paper proposes a pipeline that allows the prediction of large-model performance using small-scale experiments.

3. Applications in Continual Training: The framework also provides insights for maintaining performance in continual pretraining by dynamically adjusting mixture proportions, thus preventing catastrophic forgetting.

**Strengths:**

1. The paper is well-written and easy to understand.

2. The research topic on pre-training data mixture is timely and important for the LLM community.

3. The visualization of the model weights t_{ij} in Figure 5 is interesting because it shows the nuanced interaction between the validation domain and training domain proportions.

**Weaknesses:**

1. The validness of the underlying domain-independent assumption. Similar to other recent papers that model the data mixture problem using scaling law like [1], there is also an implicit assumption of the proposed method: different domain proportions are independent. In Eq. (1), no feature interaction term like r_i*r_j models the interactions between domain proportions. The loss is modeled by an exponential function over the **linear combination** of the proportions. We would like to understand how each domain affects every other domain, this interaction might be highly complex and not local. I would like to further clarify that the feature/proportions interaction mentioned here is different from those shown in Figure 5.

2. Missing baselines. There does not seem to be a sufficient comparison between the proposed method and other data selection methods, e.g. comparing with other token-level and sample-level automatic data selection methods [2, 3, 4].

3. Missing downstream performances. Only perplexity is evaluated, which is not sufficient to demonstrate the effectiveness of the proposed method.

[1] BiMix: Bivariate Data Mixing Law for Language Model Pretraining

[2] Perplexed by Perplexity: Perplexity-Based Data Pruning With Small Reference Models

[3] DoGE: Domain Reweighting with Generalization Estimation

[4] DoReMi: Optimizing Data Mixtures Speeds Up Language Model Pretraining

**Questions:**

See the Weaknesses.

I am open to raising my score if the authors address the above Weaknesses.

---

> ### Author Response · Authors · 2024-11-19
> **Responses to Reviewer uTnD (1/2)**
>
> Dear Reviewer uTuD,
>
> Thank you for your constructive comments. We summarize the questions you concern about and answer them as follows.
>
> > Q1: The validness of the underlying domain-independent assumption. No feature interaction term like r_i*r_j models the interactions between domain proportions
>
> We would like to clarify that **we have not made domain-independent assumptions**.
>
> **Theoretically**, Eqn. 1 in fact includes the feature interaction term like $r_ir_j$ as you mention given the linear combination serves as an exponent. Specifically, by Taylor expansion, we have
> $$L_i(r_{1\dots M}) = c_i + k_i\exp{\left(\sum_{j=1}^M{t_{ij}r_j}\right)}=c_i+k_i\left(1 + \sum_{j=1}^M{t_{ij}r_j}+\frac{1}{2}\sum_{j=1}^{M}\sum_{k=1}^Mt_{ij}t_{ik}r_jr_k+o^2\right),$$
>
> where there exists cross terms $r_jr_k (j\not=k)$ modeling the interaction between different domains as you mention.
>
> **Empirically**, we conduct a synthetic experiment, where the two experimented domains are (1) 50% Wikipedia + 50% CommomCrawl and (2) 50% Wikipedia + 50% ArXiv. The two domains are obviously correlated with the 50% wiki data. In this case, **as shown in Fig. 16 in Appendix F.2, Eqn. 1 can still model the relationship between mixture proportion and validation losses**. Here are the specific results.
>
> |Proportion of "50% Wikipedia + 50% CommomCrawl"|0.375|0.4625|0.5|0.5375|0.625|
> |-|-|-|-|-|-|
> |Observed loss on "50% Wikipedia + 50% CommomCrawl"|3.931|3.854|3.823|3.741|3.685|
> |Predicted loss on "50% Wikipedia + 50% CommomCrawl"|3.936|3.844|3.805|3.767|3.679|
> |Absolute Error|0.005|0.010|0.018|0.026|0.006|
> |-|-|-|-|-|-|
> |Observed loss on "50% Wikipedia + 50% ArXiv"|2.066|2.095|2.134|2.133|2.143|
> Predicted loss on "50% Wikipedia + 50% ArXiv"|2.073|2.101|2.114|2.126|2.155|
> |Absolute Error|0.007|0.006|0.020|0.007|0.012|
>
> We hope the above discussion helps explain the efficacy of our formula better. We are grateful for your comments and include these discussions in Appendix F.2 to improve the clarity of our manuscript.
>
> ---
>
> > Q2: Missing baselines. There does not seem to be a sufficient comparison between the proposed method and other data selection methods.
>
> We have compared our method to DoGE[1] and DoReMi[2], which you mention, in Fig. 9.
> For a more comprehensive comparison, as also suggested by other reviewers, we update the results of DoGE and DoReMi in Fig. 9 with more strictly optimized settings.
>
> For DoGE, we further include their OOD version algorithm that aligns more with our setting. For DoReMi, we include another data mixture that is adapted from the mixture it optimizes specifically for the Pile.
>
> **We include the details in Appendix F.4 in our updated manuscript. Here are the results of training 1B models for 100B tokens, which shows the data mixture our data mixing laws provide is still optimal.**
>
> $~$
>
> *Tab. 1* Comparison of different data mixture optimization methods. Default: the mixture originally used by LLaMa. DoGE (Universal): DoGE for universal generalization, which optimizes on a validation set i.i.d. to training data, we obtain the mixture from [1]. DoGE (OOD): DoGE optimized for a given validation data. DoReMi (RedPajama): DoReMi optimized for RedPajama. DoReMi (Pile): DoReMi mixture adapted from the results of optimizing on the Pile.
>
> ||Default|DoGE (Universal)|DoGE (OOD)|DoReMi (RedPajama)|DoReMi (Pile)|Ours|
> |-|-|-|-|-|-|-|
> |Validation Loss|2.795509|2.811973|2.792664|2.789486|2.840071|**2.757676**|
>
> $~$
>
> For another paper you mention [3], **it targets on sample selection problem instead of data mixing**, which are two orthogonal problems. It can be applied together with our as well as other data mixing methods.
>
> [1] DOGE : Domain Reweighting with Generalization Estimation
>
> [2] DoReMi: Optimizing Data Mixtures Speeds Up Language Model Pretraining
>
> [3] Perplexed by Perplexity: Perplexity-Based Data Pruning With Small Reference Models

---

> > ### Author Response · Authors · 2024-11-19
> > **Responses to Reviewer uTnD (2/2)**
> >
> > > Q3: Missing downstream performances. Only perplexity is evaluated, which is not sufficient to demonstrate the effectiveness of the proposed method.
> >
> > Thank you for your suggestions. Here we include the downstream performance of models trained on different mixtures. The optimal is in bold while the second and third are underlined. Our method achieves the best performance on Winogrande and BoolQ, while maintaining highly competitive performance on other tasks, leading to the highest average score across all tasks.
> >
> > ||Winogrande|Hellaswag|LAMBADA|BoolQ|Average|
> > |-|-|-|-|-|-|
> > |Default|$\underline{51.14}$|$36.90$|$\underline{47.12}$|$61.96$|$\underline{49.28}$|
> > |DoGE (Universal)|$50.35$|$36.35$|$46.59$|$61.96$|$48.80$|
> > |DoGE (OOD)|$50.28$|$\underline{37.56}$|$46.56$|$60.18$|$48.64$|
> > |DoReMi (RedPajama)|$50.59$|$35.48$|$44.36$|$\underline{62.08}$|$48.13$|
> > |DoReMi (Pile)|49.72|$\textbf{38.28}$|$\textbf{47.23}$|$\underline{62.11}$|$\underline{49.33}$|
> > |Ours|$\textbf{51.62}$|$\underline{36.94}$|$\underline{47.04}$|$\textbf{62.23}$|$\textbf{49.46}$|
> >
> > ---
> >
> > We hope our discussion, together with the supplemented experiment results, helps present our work more comprehensively. Please let us know if any questions remain and we are happy to discuss and address them. Looking forward to your feedback.

---

> ### Comment · Reviewer_uTnD · 2024-11-20
>
> Thanks for the answers, especially the Taylor expansion part.
> This mitigates my main concern regarding the modeling of domain interactions.
> The additional experiments for more baselines and downstream tasks also make some sense to me.
> Therefore, I have increased my score to 6.

---

> > ### Author Response · Authors · 2024-11-21
> >
> > We sincerely appreciate your thoughtful review and the detailed consideration you've given to our work.
> > We are gratified that our clarification resolves your primary concern and the supplementary experiments substantiate our claims.
> >
> > We remain committed to refining our manuscript based on your invaluable advice.

---

### Official Review · Reviewer_1wD6 · 2024-11-05

**Soundness:** 2
**Presentation:** 4
**Contribution:** 3
**Rating:** 6
**Confidence:** 4

**Summary:**

In this paper, the authors carefully describe the steps they took to arrive at so-called data mixing law which is a simple formula that predicts the validation loss (potentially of arbitrary mix of diverse domains) of a model trained on a fixed data mixing ratio. This law has O(n_domain x n_domain + n_domain) parameters. Although there are a few arbitrary decisions here and there (e.g., choosing M4 over M1) and the validation (of the method and law) is limited to just one particular setup (training a language model on a relatively general set of domains), i enjoyed reading the authors’ thought process arriving at the proposed “law”.

**Strengths:**

I very much enjoyed reading how the authors present their thought process leading to the proposed law. Each step was reasonable, and some of the assumptions the authors put forward made sense (although it is never possible to know whether it’s post hoc or ad hoc by reading the manuscript alone.) Because of clear presentation of the steps, I was reasonably convinced of the utility of the proposed law for figuring out the data mixing ratio, despite its limited empirical validation. Furthermore, it was good to see that the authors showed that this law can be used in conjunction with so-called scaling law to improve large-scale training based on the mixing law from smaller scale training.

**Weaknesses:**

There are two main weaknesses. First, any such law is by nature empirical and thereby must be validated quite extensively with terms of their predictive power, before they can be established as the law. The experiments in the paper are reasonable, but are conducted in exactly one setup. Although the authors consider continual pretraining as a separate setup, i don’t believe it is separate enough to show that this law generalizes beyond this particular setup. I would’ve even appreciated some synthetic experiments, as long as some diversity was introduced. This is the major weakness, since the authors claim it to be the “law” of data mixing.

The second weakness is whether this particular form is the right form. As the authors pointed out, one can think of this as a particular form of a multilayer perceptron. A natural question is then whether this particular form is the one we should prefer. For instance, why do we want to choose the form out of M1, M2, M3 and M4; can’t we simply build a much simpler MLP and fit the MLP directly? If the goal is to use this so-called law to improve the efficiency of training, why would we care about a particular form of this law? Instead, we would much rather care about the predictive power of any predictor fitted on a large number of training runs.

**Questions:**

I would appreciate the authors’ responses as well as plans to address two weaknesses I have described above.

---

> ### Author Response · Authors · 2024-11-19
> **Responses to Reviewer 1wD6**
>
> Dear Reviewer 1wD6,
>
> Thank you for your insightful comments. We are pleased that you find our analysis and thought process enjoyable to read and is convincing. We summarize your concerns and try addressing them below.
>
> > Q1: The law is empirical and must be validated quite extensively in terms of their predictive power. The experiments in the paper are reasonable, but are conducted in exactly one setup.
>
> We appreciate your recognition of the thought process we present and suggest it reasonably convinding. We do understand your concerns about limited empirical setups. To further confirm the generality of data mixing laws, we include additional experiments on more data domains.
>
> In specific, we further experiment on Wikipedia, ArXiv, and StackExchange, which are three domains different from our experiments in Sec. 3.2. We train 70M models for 10k steps on different mixtures and fit Eqn. 1 with 7 mixtures and validate on the other 3 mixtures. We visualize our results in Appendix F.1 and present the validation accuracy as follows. **The result confirms data mixing laws still hold in these domains.** We hope this can further support the generality of our data mixing laws.
>
> $~$
>
> Tab. 1 Loss prediction errors of data mixing laws on the mixture of Wikipedia, ArXiv, and StackExchange.
>
> |Training Mixture Proportion (Wikipedida:ArXiv:StackExchange)|0.25:0.25:0.5|0.25:0.5:0.25|0.375:0.5:0.125|
> |-|-|-|-|
> |Prediction on Wikipedia Loss|4.978 |4.987|4.860|
> |Observation on  Wikipedia Loss |4.994  |5.024|4.893|
> |Absolute Prediction Error on Wikipedia Loss |0.016|0.038|0.033|
> |-|-|-|-|
> |Prediction on ArXiv Loss |2.077|1.983|1.989|
> |Observation on  ArXiv Loss |2.068|1.996|2.017|
> |Absolute Prediction Error on ArXiv Loss |0.009|0.013|0.028|
> |-|-|-|-|
> |Prediction on StackExchange Loss |2.223|2.272|2.378|
> |Observation on  StackExchange Loss |2.179|2.208|2.408|
> |Absolute Prediction Error on StackExchange Loss |0.044|0.064|0.030|
>
> ---
>
> > Q2: Why would we care about a particular form of this law?
>
> Thanks for insightful discussion. We believe many function forms (e.g., polynomials and MLP) can predict the model performance regarding data mixing well as we attribute the prediction accuracy of data mixing laws to that it is actually predicting through interpolation.
>
> Despite many other potential alternatives, **a specific form of data mixing laws provide the following benefits**:
>
> (1) Simplicity. Including fewer parameters helps using fewer samples to fit the law. It is easier to analyze. For example, our exponential law indicates that the benefit of increasing the proportion of a domain decreases exponentially, which implies the benefits of using a balanced mixture.
>
> (2) Interpretability. The function form of data mixing laws can provide some interpretability on the parameters. For instance, we show the interaction between different domains (Fig. 5) with $t_ij$ in our data mixing law.
>
>
> For predictive powers, we have also experimented with fitting with MLP directly but does not find an obvious difference in the accuracy compared our empirical formulas. Under the setting of Fig. 7, the validation error with data mixing laws (Eqn. 8) and an MLP are 0.01713 and 0.01727, respectively.
>
>
>
> ---
> We hope our response helps clarify our ideas and consolidate our work better. We look for your feedback.

---

### Official Review · Reviewer_c3fa · 2024-11-09

**Soundness:** 3
**Presentation:** 3
**Contribution:** 4
**Rating:** 8
**Confidence:** 5

**Summary:**

It is important to determine the best mixture proportions over a set of domains (i.e., code, web text, for pre-training). However, naive approaches require manual tuning over the search space, and recent algorithms that learn the proportions are not well-understood. Instead, this paper discovers a log-linear "mixing law" that relates mixture proportions to loss. Experiments are done over 2, 3, 7 domains (Pile, RedPajama) to confirm that this mixing law is accurate. This mixing law provides a way of determining the optimal data mixture in a principled manner: conduct multiple training runs over various mixtures, fit the parameters of the mixing law, and use those to compute the proportions that minimize the desired loss. One drawback of naively using this approach is that this requires many full training runs. The paper thus proposes layering on training-step and model-size scaling laws so that the optimal mixture for some target model size and target number of training steps can be inferred by training on various mixtures for smaller models and fewer number of steps. The paper also discusses various ways this approach can be used: it can be applied when we are given a validation dataset that is disjoint from training domains, and it can be applied in the continual pre-training setting too. Empirical results show that conducting shortened runs on smaller models allows them to produce a mixture over RedPajama that attains low validation loss on the Pile faster than the default mixture.

**Strengths:**

Originality:
- The paper departs from other data mixing works by presenting the following novel methodology: let's first focus on learning the parametric form of the relationship between proportions and loss, and then use this to find the proportions that minimize the predicted loss. The idea is inspired by work on scaling laws, but mixing laws are quite different. They must capture different factors, such as cross-domain interactions, and can be multi-dimensional depending on the number of domains.

Quality:
- Beyond just presenting the mixing laws, which are interesting as a scientific artifact, the paper does a good job of discussing practical considerations (training budget) as well as insights from fitting the laws, such as the sparsity of the interaction matrix.
- The paper explains how mixing laws are broadly applicable, presenting practical settings that are not just limited to minimizing loss on the training domains (which e.g., DoReMi focuses on). In particular, the paper considers when we have a separate validation dataset, and when we wish to continually pre-train a model to learn some new domain without forgetting the original one.

Clarity:
- Paper was well-structured in starting with results on the simplest mixing law and then gradually moving towards more practical settings.

Significance:
- The paper provides a feasible method of doing data mixing that can be much cheaper than brute-force searching---which is often the convention when training LLMs---and is more predictable than recent algorithmic approaches. It captures the intuition of the practical trial-and-error process that practitioners would make when tuning mixtures, and formalizes it into a rigorous procedure.

**Weaknesses:**

Clarity:
- Minor typos: "the implementation details." on L328. Figure 6 has no x-axis. In L409, the second L(S) should be L(N). In L473 "date" should be "data".
- What does resampling mean in L456? The procedure here does not appear to match up with the algorithm since you only mention training 70M models.
- My main clarity issue is in Algorithm 1. Presented as is, it is very difficult for someone to implement. What are the (x, y)'s you need to store in each loop and fit? What are the exact scaling laws? While L(S) and L(N) are defined in L409, this does not typecheck with the algorithm using only L(N, S, r)s.

Quality:
- Evaluation of DoGE and DoReMi: the mixtures taken from their paper are for the "universal generalization" setting; for DoGE these are the learned proportions that aim to minimize the average validation loss across domains, not a disjoint validation loss. For a fair comparison, you would need to use the OOD version of DoGE (their algorithm 2). Furthermore, it is unclear how much compute is used by your method vs DoGE vs DoReMi; the compute should be normalized to make a fair comparison, especially since DoGE and DoReMi require 1 extra and 2 extra training runs while this method has parameters scaling in the number of domains.
- It is unclear if the proposed mixing law holds on other domains. The 2 domain experiment is only done on Pile-CC and Github, and the 3 domain experiment is only done on Pile-CC, Github, and Books3. Is it possible that some property of these domains gives way to the linearity in the mixing law? It would also be interesting to explore if this mixing law held on tasks (i.e. in instruction-tuning).
- Table 15 in the appendix suggests to me that the mixing laws are not that accurate for the nested approach.

**Questions:**

1. Can you explain Algorithm 1's nested structure more? It would be great if there was a longer version in the appendix that could stand on its own.
2. Can you comment on the mixing law fitting well in other settings (such as different subsets of RedPajama/Pile, or a new dataset overall)?
3. Is there a better way to evaluate DoGE and DoReMi in Figure 9 than using the mixtures reported in the DoGE paper? What do the DoGE mixtures look like in the OOD setting, which would account for the shift between RedPajama and the Pile?

Overall I think this is a very nice paper. It could benefit from more clarity in describing the algorithm and a fairer evaluation of the method.

---

> ### Author Response · Authors · 2024-11-19
> **Responses to Reviewer c3fa (1/3)**
>
> Dear Reviewer c3fa,
>
> We sincerely appreciate your recognition of our work, with highlights in novelty, quality, clarity, and significance. We would also like to acknowledge the thorough and constructive feedback provided which help strengthen our  work, which we summarize and respond as follows.
>
> ---
>
> > Q1: Clarity issues on Algorithm 1.
>
> Algorithm 1 demonstrates how we predict language modeling performance across target model sizes, training steps, and data mixtures, as illustrated in Fig. 1. Formally, our goal is to predict the performance of a model with scale $N_{target}$ trained for $S_{target}$ steps on data mixture $r_{target}$.
>
> The process begins with $N$ different data mixtures for fitting, where for each mixture, we train $K$ models of different scales for $S_0$ steps.
> 1. For each trained model, we first fit $L(S)$ using its losses from the first $S_0$ steps to predict its performance at $S_{target}$.
> 2. Then, for each mixture, we use the predicted performance at $S_{target}$ across different model scales to fit $L(N)$ and predict the performance of a model with $N_{target}$.
> 3. Finally, we use these predictions from $N_{target}$ models trained for $S_{target}$ steps on different mixtures to fit data mixing laws and estimate model performance for any unseen data mixture at the given scale.
>
> We appreciate your suggestion for making the laws used each step more explicit and accordingly updated the notations in Alg. 1 with $L(S)$ and $L(N)$ to improve clarity.
>
> ---
>
> > Q2: What does resampling mean in L456?
>
> Regarding L456 in the original manuscript (L458 in the updated version), **we detail our process for selecting data mixtures to run Algorithm 1**.
> Since different fitting mixtures can yield varying prediction accuracies on data mixing laws, we use a cross-validation approach with the smallest models (which are computationally efficient) to identify effective data mixtures which helps fit a data mixing law that predict well.
>
> Our selection process involves: (1) initially training on 40 diverse mixtures, (2) sampling 20 mixtures and validating their predictions on the remaining 20, and (3) repeating step (2) and selecting the set of 20 mixtures that minimizes prediction errors on the validation set after fitting.
>
> We then use the selected mixtures to run Algorithm 1 to predict language modeling performance of large scale training on different data mixtures.

---

> ### Author Response · Authors · 2024-11-19
> **Responses to Reviewer c3fa (2/3)**
>
> > Q3: More comparison to the OOD setting of DoGE and DoReMi.
>
> Thanks for your advice to make our comparison more comprehensive. **We update further results of DoGE and DoReMi here along with the experimental setup and results in Sec. 4.2 with details in Appendix F.4.**
>
> For DoGE, we follow the original paper and train a small proxy model (160M) for 10k steps with their OOD version algorithm and obtain the data mixture as follows. We include the optimization process in Fig. 19.
>
> |Domain|CC|Books|C4|GitHub|StackExchange|Wikipedia|ArXiv|
> |-|-|-|-|-|-|-|-|
> |Proportion|0.375|0.1953125|0.29296875|0.0703125|0.1171875|0.00390625|0.12109375|
>
> For DoReMi, our original experiments were conducted with optimization on RedPajama training data. As the reviewers indicate, this may not align optimally with validation on the Pile given that DoReMi serves as a universal optimization method without awareness of validation data.
>
> To provide a more comprehensive evaluation, we leveraged the domain overlap between RedPajama and the Pile datasets to adapt the optimized mixture for the Pile, provided by the DoReMi paper, to our setting. We map the mixture proportions as follows. (1) We borrow the proportion of GitHub, StackExchange, Wikipedia, and ArXiv. (2) We let CommonCrawl and C4 share the proportion of Pile-CC and (3) let the proportion of Books as the sum of BookCorpus2 and Books3. We normalized the proportion to ensure they sum up to 1 and train with RedPajama with the normalized mixture proportions. The specific data mixture is as follows.
>
> |Domain|CC|Books|C4|GitHub|StackExchange|Wikipedia|ArXiv|
> |-|-|-|-|-|-|-|-|
> |Proportion (before normalization)|0.6057 / 2 (half of Pile-CC)|0.0061+0.0224 (Books3+BookCorpus2)|0.6057/2 (half of Pile-CC)|0.0179|0.0153|0.0699|0.0036|
> |Proportion (normalized)|0.41015625|0.0390625|0.41015625|0.0234375|0.01953125|0.09375|0.00390625|
>
> The results are as follows, which shows that **data mixture given by ours remains optimal**.
>
> $~$
>
> *Tab. 1* Comparison of different data mixture optimization methods. Default: the mixture originally used by LLaMa. DoGE (Universal): DoGE for universal generalization, which optimizes on a validation set i.i.d. to training data, we obtain the mixture from [1]. DoGE (OOD): DoGE optimized for a given validation data. DoReMi (RedPajama): DoReMi optimized for RedPajama. DoReMi (Pile): DoReMi mixture adapted from the results of optimizing on the Pile.
>
> | |Default|DoGE (Universal)|DoGE (OOD)|DoReMi (RedPajama)|DoReMi (Pile)|Ours|
> |-|-|-|-|-|-|-|
> |Validation Loss|2.795509|2.811973|2.792664|2.789486|2.840071|**2.757676**|
>
> [1] DOGE : Domain Reweighting with Generalization Estimation
>
> [2] DoReMi: Optimizing Data Mixtures Speeds Up Language Model Pretraining

---

> > ### Author Response · Authors · 2024-11-19
> > **Responses to Reviewer c3fa (3/3)**
> >
> > > Q4: It is unclear if the proposed mixing law holds on other domains. Can you comment on the mixing law fitting well in other settings? Is it possible that some property of these domains gives way to the linearity in the mixing law?
> >
> > Thank you for your suggestions to include results on more domains to show the generality of data mixing laws.
> >
> > **We conducted additional experiments across three domains different from those presented in our orignal manuscript: Wikipedia, arXiv, and StackExchange.** We train 70M models for 10k steps on different mixtures and fit Eqn. 1 with 7 mixtures and validate on the other 3 mixtures. **We update the results in Appendix F.1 and present the validation accuracy as follows, which confirms the efficacy of data mxing laws on these domains.**
> >
> > $~$
> >
> > |Training Mixture Proportion (Wikipedida:ArXiv:StackExchange)|0.25:0.25:0.5|0.25:0.5:0.25|0.375:0.5:0.125|
> > |-|-|-|-|
> > |Prediction on Wikipedia Loss|4.978 |4.987|4.860|
> > |Observation on  Wikipedia Loss |4.994  |5.024|4.893|
> > |Absolute Prediction Error on Wikipedia Loss |0.016|0.038|0.033|
> > |-|-|-|-|
> > |Prediction on ArXiv Loss |2.077|1.983|1.989|
> > |Observation on  ArXiv Loss |2.068|1.996|2.017|
> > |Absolute Prediction Error on ArXiv Loss |0.009|0.013|0.028|
> > |-|-|-|-|
> > |Prediction on StackExchange Loss |2.223|2.272|2.378|
> > |Observation on  StackExchange Loss |2.179|2.208|2.408|
> > |Absolute Prediction Error on StackExchange Loss |0.044|0.064|0.030|
> >
> > For the property that makes data mixing law reliable, we consider it as the **interpolation nature of the problem of predicting the performance of data mixing, which makes prediction relatively easy**. While other functional forms might also effectively capture this interpolation behavior, our proposed exponential function is simple and empirically capable.
> >
> > ---
> >
> > > Q5: It would also be interesting to explore if this mixing law held on tasks (i.e. in instruction-tuning).
> >
> > Thank you for your invaluable advice.  Extending performance prediction towards downstream tasks is an exciting topic to explore. We also notice many recent works making excellent progress in this field [1, 2, 3]. Given the interpolation nature of data mixing prediction, we feel optimistic about prediction on downstream tasks regarding data mixture, which we plan to explore in the near future.
> >
> > [1] Observational Scaling Laws and the Predictability of Language Model Performance
> >
> > [2] Predicting Emergent Abilities with Infinite Resolution Evaluation
> >
> > [3] Scaling laws for neural machine translation
> >
> > ---
> >
> > > Q6: Fig. 20 ( in the updated version) suggests the nested approach is not that accurate.  It is unclear how much compute is used by the method.
> >
> > Both accuracy and computation requirement in our approach highly relates to how much scaling laws accurately extrapolate.
> >
> > We attribute the deviation of predictions in Fig. 20 to the cumulative effects of errors from multiple law. As Fig. 6, Fig. 7 and results in Appendix D.2 indicate, scaling laws and data mixing laws both introduce errors of around 0.02 which compound in the final predition. Nevetheless, we find these errors do not changed the ranking of different data mixture.
> >
> > Regarding the computation required, it is the number of fitting mixture multiplied by the costs to fit scaling laws for each mixture. Typically, we fit with 20 mixtures. The state-of-the-art scaling laws prediction only require computation budget of 1/100 - 1/1000 of the final experiments[1,2]. Thus, in total, this only require 1/5 to 1/50 extra computation. While at the time we finished our initial manuscript, admittedly, we do not have sufficient computation resources to extensively validate the scaling prediction at such large scales, our latest experiment shows that L(S) itself is able to achieve at least 6.25x scaling and L(N) 70x scaling, **with results we updated in Appendix F.3**.
> >
> > Given the scaling prediction remains a developing research area[3,4,5], the accuracy and efficiency of our method can be further improved with more investigations into the practice of scaling-law prediction, which would be a focus of our future work.
> >
> > [1] GPT4 Technical Report
> >
> > [2] DeepSeek LLM Scaling Open-Source Language Models with Longtermism
> >
> > [3] Unraveling the Mystery of Scaling Laws: Part I
> >
> > [4] Scaling Laws with Learning Rate Annealing
> >
> > [5] A Hitchhiker’s Guide to Scaling Law Estimation
> >
> > ---
> >
> > Overall, we are grateful for your valuable comments and praise on our work. We are delighted to include your suggestion in our updated manuscript to further improve the quality.

---

### Meta-Review · Area_Chair_Vi3t · 2024-12-22

**Metareview:**

The authors propose to estimate optimal training data mixture proportions by training a series of smaller models with varying mixture proportions and then fitting a simple model to relate those proportions to performance. They show that this leads to more efficient training (as a function of the number of training tokens) compared to baselines. Training data optimization is an important problem as large training runs are expensive; the approach here is simple, natural, and well-executed. Reviewers were unanimous in recommending acceptance and I recommend acceptance as well.

I encourage the authors to take the reviewers' feedback into account. For example, as Reviewer c3fa mentions, it would be helpful to discuss the relative computational costs of these different data mixture optimization methods, since the amount of compute spent on such methods could instead be used to train on a (possibly less optimal) data mixture for longer. In addition, please incorporate the experiments on other domains into the main text. Finally, I suggest that the authors compare with related concurrent work: https://arxiv.org/abs/2407.01492.

**Additional Comments On Reviewer Discussion:**

There were some follow-up questions asking for additional ablations, which the authors answered satisfactorily.

---

### Decision · Program_Chairs · 2025-01-22

Accept (Poster)